# A RAD51 assay feasible in routine tumor samples calls PARP inhibitor response beyond BRCA mutation

Marta Castroviejo-Bermejo[1], Cristina Cruz[1,2,3], Alba Llop-Guevara[1], Sara Gutiérrez-Enríquez[4], Mandy Ducy[5,6,7], Yasir Hussein Ibrahim[1], Albert Gris-Oliver[1], Benedetta Pellegrino[1,8], Alejandra Bruna[9], Marta Guzmán[1], Olga Rodríguez[1], Judit Grueso[1], Sandra Bonache[4], Alejandro Moles-Fernández[4], Guillermo Villacampa[10], Cristina Viaplana[10], Patricia Gómez[3,11], Marc Vidal[3,11], Vicente Peg[12,13], Xavier Serres-Créixams[14], Graham Dellaire[15], Jacques Simard[7], Paolo Nuciforo[13,16], Isabel T Rubio[13,17], Rodrigo Dientsmann[10], J Carl Barrett[18], Carlos Caldas[9,19], José Baselga[20,21], Cristina Saura[3,11], Javier Cortés[13,22,23], Olivier Déas[24], Jos Jonkers[25] ⓘD, Jean-Yves Masson[5,6], Stefano Cairo[24], Jean-Gabriel Judde[24], Mark J O'Connor[26], Orland Díez[4,27], Judith Balmaña[2,3,*] ⓘD & Violeta Serra[1,13,**] ⓘD

## Abstract

Poly(ADP-ribose) polymerase (PARP) inhibitors (PARPi) are effective in cancers with defective homologous recombination DNA repair (HRR), including BRCA1/2-related cancers. A test to identify additional HRR-deficient tumors will help to extend their use in new indications. We evaluated the activity of the PARPi olaparib in patient-derived tumor xenografts (PDXs) from breast cancer (BC) patients and investigated mechanisms of sensitivity through exome sequencing, *BRCA1* promoter methylation analysis, and immunostaining of HRR proteins, including RAD51 nuclear foci. In an independent BC PDX panel, the predictive capacity of the RAD51 score and the homologous recombination deficiency (HRD) score were compared. To examine the clinical feasibility of the RAD51 assay, we scored archival breast tumor samples, including PALB2-related hereditary cancers. The RAD51 score was highly discriminative of PARPi sensitivity versus PARPi resistance in BC PDXs and outperformed the genomic test. In clinical samples, all PALB2-related tumors were classified as HRR-deficient by the RAD51 score. The functional biomarker RAD51 enables the identification of PARPi-sensitive BC and broadens the population who may benefit from this therapy beyond BRCA1/2-related cancers.

**Keywords** *BRCA1*; homologous recombination; *PALB2*; PARP inhibitors; RAD51

**Subject Categories** Biomarkers & Diagnostic Imaging; Cancer; Pharmacology & Drug Discovery

## Introduction

Poly(ADP-ribose) polymerases (PARPs) are enzymes with diverse functions, including repair of DNA single-strand breaks (SSBs). PARP inhibition not only impairs SSB repair but also results in PARP trapping onto DNA with subsequent stalling of replication forks (Plummer, 2006; Helleday, 2011; Murai *et al*, 2012; Lord & Ashworth, 2017). Both effects contribute to the formation of DNA double-strand breaks (DSBs) that, in replicated areas of the genome, are repaired by homologous recombination repair (HRR), a conservative mechanism for error-free repair of DNA damage (Saredi *et al*, 2016; Pellegrino *et al*, 2017). Cells with defects in HRR including those with deleterious variants in *BRCA1* or *BRCA2* (*BRCA1/2*) genes are particularly sensitive to PARP inhibitors (PARPi; Bryant *et al*, 2005; Farmer *et al*, 2005; Rottenberg *et al*, 2008), which prompted the clinical development of PARPi as anticancer therapies (Fong *et al*, 2009; Audeh *et al*, 2010; Tutt *et al*, 2010). In breast cancer (BC), the efficacy results of the PARPi olaparib (Lynparza®) in metastatic patients carrying a germline *BRCA1/2* (gBRCA) pathogenic variant have led to its recent approval by the Food and Drug Administration (Robson *et al*, 2017). PARPi have also shown preclinical and clinical activity beyond gBRCA in ovarian and prostate cancer (McCabe *et al*, 2006; Kaufman *et al*, 2014; Mateo *et al*, 2015; Mirza *et al*, 2016; Coleman *et al*, 2017; Pujade-Lauraine *et al*, 2017).

Similarly, the use of PARPi could be extended beyond gBRCA to a wider group of BC patients harboring dysfunctional HRR. For example, the clinical and molecular similarities between *BRCA1*-associated tumors and a subset of triple negative BCs (TNBC) led to postulate that the latter may also have defects in HRR (Turner *et al*, 2004). In such cases, HRR deficiency can be explained by epigenetic

1–27  The list of affiliations appears at the end of this article
*Corresponding author. Tel: +34 93 2746000; Fax: +34 93 4894212; E-mail: jbalmana@vhio.net
**Corresponding author. Tel: +34 93 2543450; Fax: +34 93 4894212; E-mail: vserra@vhio.net

silencing of *BRCA1/2* or the genetic inactivation of several other HRR-related genes such as *ATM, ATR, CHEK1, CHEK2, PALB2,* and the *FANC* family genes (Konstantinopoulos *et al*, 2015; Shakeri *et al*, 2016). Specifically, PALB2 is essential for BRCA2 anchorage to nuclear structures and recruitment to DSBs, acting as the link between BRCA1 and BRCA2 (Buisson & Masson, 2012; Pauty *et al*, 2014).

Despite the success of PARPi monotherapy in gBRCA BC, appropriate biomarkers are still needed for selection of non-gBRCA patients for PARPi therapy (Gelmon *et al*, 2011; Mutter *et al*, 2017). Some proposed approaches include the use of mRNA expression signatures, the analysis of genomic scars derived from defective HRR, or the individual analysis of genetic alterations in HRR-related genes (Konstantinopoulos *et al*, 2010; Abkevich *et al*, 2012; Wagle *et al*, 2012; Watkins *et al*, 2014; Davies *et al*, 2017; Polak *et al*, 2017). A potential limitation of these approaches is the lack of specificity in HRR-altered tumors that have restored the HRR function (Watkins *et al*, 2014; Konstantinopoulos *et al*, 2015). Other approaches entail the quantification of *BRCA1* promoter hypermethylation, *BRCA1* mRNA expression, or the detection of the HRR protein RAD51 forming nuclear foci after DNA damage, as surrogate of HRR functionality (Graeser *et al*, 2010; Naipal *et al*, 2014; ter Brugge *et al*, 2016). In these sense, we showed that, in gBRCA tumors, RAD51 foci could be detected in untreated samples and correlated with PARPi resistance regardless of the underlying mechanism restoring HRR function (Cruz *et al*, 2018).

In this work, we analyzed five HRR biomarkers (genetic alterations in HRR-related genes, *BRCA1* promoter methylation, *BRCA1* expression, BRCA1 foci formation, and RAD51 foci formation) and tested which one performed better to predict PARPi response. Importantly, we further show that the RAD51 assay is feasible in routine formalin-fixed paraffin-embedded (FFPE) tumor samples without prior induction of DNA damage. Scoring RAD51 allowed the identification of non-gBRCA HRR-deficient BCs with high accuracy, which may help identify a wider BC population with intrinsic sensitivity to PARPi therapy.

# Results

## Olaparib antitumor activity in a non-gBRCA BC patient-derived tumor xenograft (PDX) panel distinguishes a subset of tumors highly sensitive to PARPi

We assessed the antitumor activity of the PARPi olaparib in 18 PDX models derived from non-gBRCA BC patients (PDX cohort-1, Table EV1). Treatment with olaparib revealed antitumor activity in four PDX models as assessed by mRECIST (see Materials and Methods): complete response (CR, *n* = 2: PDX093 and PDX197) or partial response (PR, *n* = 2: PDX302 and STG201). The remaining eleven PDX models were olaparib-resistant (PD, progressive disease; Fig 1A and Appendix Table S1). Additional resistant models were generated from three of the four olaparib-sensitive PDXs after prolonged exposure and steep progression to olaparib (STG201OR, PDX093OR, and PDX302OR; Fig 1B). The fourth olaparib-sensitive model PDX197 did not grow after prolonged treatment (> 120 days). This PDX collection of 18 BC models was used to study clinically

relevant mechanisms of PARPi sensitivity and acquired resistance *in vivo*.

## HRR-related somatic alterations and PARPi sensitivity in PDX

We next investigated the presence of alterations in HRR-related genes that could explain olaparib sensitivity. As aberrant *BRCA1* promoter methylation is found in approximately 10% of sporadic breast cancers (Shakeri *et al*, 2016), we first measured the levels of epigenetic silencing of *BRCA1* and analyzed *BRCA1* expression and nuclear foci formation in PDX samples. Our approach validated previously reported *BRCA1* promoter methylation and expression results from the STG139 and STG201 models (Bruna *et al*, 2016). Results showed that three out of four olaparib-sensitive models (PDX302, STG201, and PDX197) and one olaparib-resistant model (PDX270) presented *BRCA1* promoter hypermethylation, while the remaining PDX models showed low levels of methylation (Fig 2A). In agreement, absence of *BRCA1* mRNA expression and lack of BRCA1 nuclear foci were restricted to the four models that showed *BRCA1* promoter hypermethylation (Fig 2A, larger views in Appendix Fig S1). Of note, the olaparib acquired-resistant models STG201OR and PDX302OR exhibited lower levels of *BRCA1* promoter hypermethylation in comparison with the olaparib-sensitive counterparts, and displayed *BRCA1* mRNA expression and BRCA1 nuclear foci formation (Fig 2A).

We then performed exome sequencing to detect genetic alterations in other genes related to HRR (Table EV2). We identified frameshift mutations in HRR-related genes in three models: *PALB2* and *FANCD2* in two PARPi-sensitive models (PDX093 and STG201, respectively) and *RAD54L* in one PARPi-resistant model (PDX270). In summary, neither epigenetic silencing of *BRCA1* nor the presence of HRR gene alterations fully associated with PARPi sensitivity.

We characterized the *PALB2* deleterious variant in the PARPi-sensitive PDX093, as it was heterozygous in the tumor. The specific *PALB2* mutation in PDX093 (c.886dupA, Fig EV1A) predicts a protein truncation in PALB2 lacking the C-terminus region (p.M296Nfs), as the known germline pathogenic variant c.886del in *PALB2* (Antoniou *et al*, 2014). By Western blot, the PALB2 wild-type protein was not detected in PDX093 (Fig 2B). We then examined the recruitment of PALB2 p.M296Nfs to DSB sites after laser-induced DNA damage. HeLa cells were transfected with YFP-PALB2-WT or YFP-PALB2-p.M296Nfs and microirradiated. The recruitment of YFP-PALB2 to laser-induced DNA damage sites was monitored for 16 min. This assay allowed us to observe that PALB2 p.M296Nfs mutant protein was not properly recruited (Fig 2C and Movies EV1 and EV2). We next studied the effect of PALB2 p.M296Nfs mutant protein on HRR capacity using a Cas9/mClover-LMNA HRR assay in U2OS cells. The Cas9/mClover-LMNA assay measures the HRR-dependent insertion of a *mClover*-containing cassette into Cas9-generated DSBs in the *LMNA* gene, resulting in a *mClover-LMNA* fusion gene encoding green lamin A/C. HRR can be monitored by looking at mClover-positive cells. While wild-type PALB2 partly complemented PALB2 siRNA-treated cells, PALB2 p.M296Nfs mutant did not rescue HRR capacity (Fig 2D). Furthermore, overexpression of PALB2 p.M296Nfs mutant led to a two-fold reduction in mClover-positive cells, demonstrating that PALB2 p.M296Nfs leads to HRR deficiency despite the presence

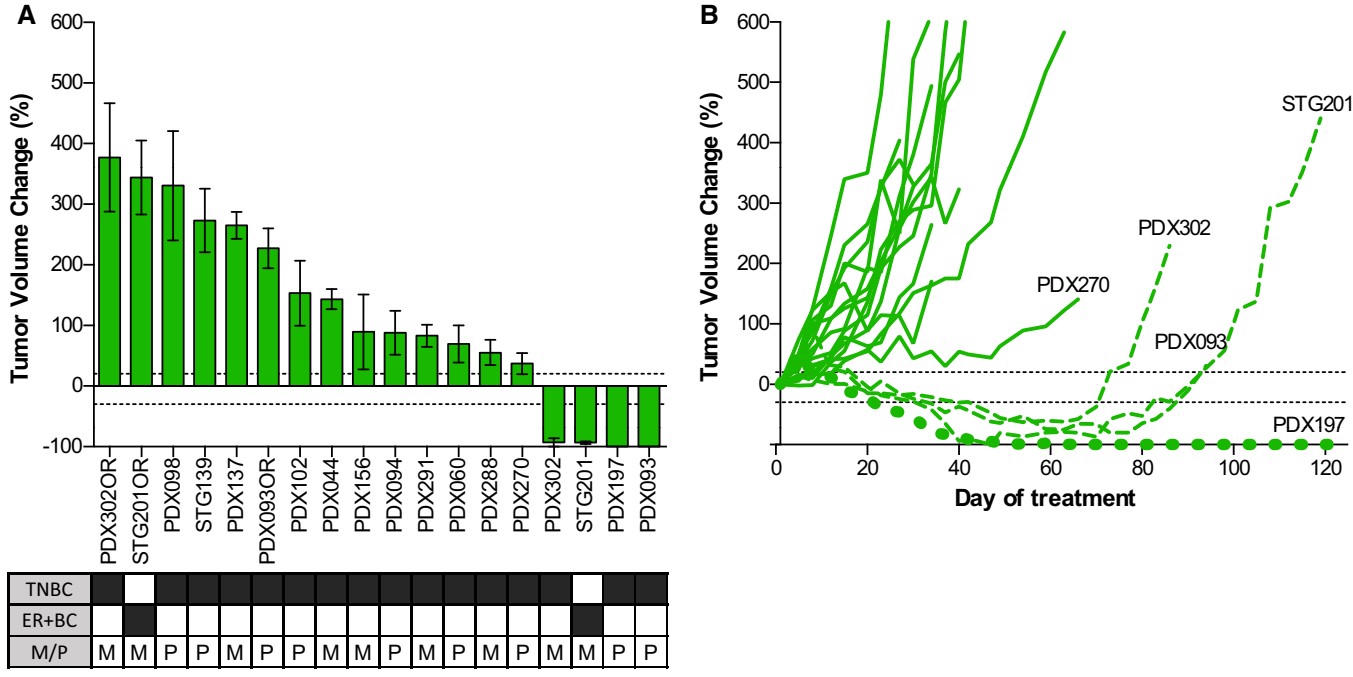

**Figure 1.  The antitumor activity of olaparib in PDXs identifies a subset of PARPi-sensitive tumors.**

A   Waterfall plot showing the percentage of tumor volume change in olaparib-treated tumors compared to the tumor volume on day 1. +20% and −30% are marked by dotted lines to indicate the range of PR, SD, and PD. The box underneath summarizes different characteristics of each model and the clinical context at the moment of PDX implantation. Black boxes indicate the presence of the phenotype. TNBC, triple negative breast cancer; ER+BC, estrogen receptor positive breast cancer; P, primary; M, metastasis. Error bars indicate SEM from independent tumors ($n \geq 3$).

B   Graph showing the percentage of tumor volume change during olaparib treatment in PDXs from cohort-1. Olaparib-sensitive models are represented with discontinuous lines. Acquisition of PARPi resistance in PDX302, STG201, and PDX093 after prolonged exposure to olaparib is shown.

of endogenous wild-type PALB2, in favor of a dominant negative effect (Fig 2E; Lee *et al*, 2018). Collectively, both sequencing and functional assays indicate that HRR deficiency in PDX093 is due to the *PALB2* c.886dupA mutation.

**Lack of RAD51 nuclear foci is associated with PARPi sensitivity**

We further investigated the functional status of HRR in FFPE tumors from PDX cohort-1, scoring the percentage of RAD51-positive tumor cells in S/G2-phase of the cell cycle (geminin-positive) following the protocol shown in Fig EV1B (Graeser *et al*, 2010; Naipal *et al*, 2014). In olaparib-treated tumors, the four PARPi-sensitive models showed significantly lower percentage of RAD51-positive cells than the fourteen PARPi-resistant models ($1.25 \pm 0.25\%$ versus $66.54 \pm 2.70\%$; $P < 0.0001$; Fig 3A, larger views in Appendix Fig S2, and Fig EV1C). In agreement with BRCA1 foci formation, the acquired-resistant models STG201OR and PDX302OR showed RAD51 nuclear foci (Fig 3B, larger views in Appendix Figs S3 and S4). While previous studies had reported low levels of baseline DNA damage as a potential limitation to evaluate HRR functionality (Graeser *et al*, 2010; Naipal *et al*, 2014), RAD51 could be scored in vehicle-treated tumors ($0.75 \pm 0.48\%$ versus $47.08 \pm 3.37\%$; $P = 0.0007$) and correlated with the antitumor activity of olaparib ($P = 0.0044$; Figs 3A and EV1C and D). We then analyzed the Receiver Operating Characteristic (ROC) Curve showing that the RAD51 score displays complete discriminative capacity in predicting PARPi response (Area Under the ROC Curve (AUC) = 1, $P = 0.0030$,

specificity = 100%, sensibility = 100%). An in-depth analysis of absolute RAD51 foci quantification demonstrated that the 5 foci-per-nucleus cutoff proved to be accurate for discrimination purposes (Appendix Fig S5). We further ruled out that lack of RAD51 foci in PARPi-sensitive PDXs was due to lack of endogenous DNA damage, by quantifying the levels of phosphorylated H2AX (γ-H2AX), a marker of DSBs (Raderschall *et al*, 1999; Petermann *et al*, 2010; Fig 3C); or by low number of cells in S/G2 phase of the cell cycle (geminin-positive; Fig EV1E). The RAD51 assay was validated using a monoclonal antibody (Fig EV1F). Altogether, these data show that the RAD51 score is highly discriminative of PARPi sensitivity versus PARPi resistance in the PDX cohort-1.

**RAD51 score predicts PDX's response to PARPi in an independent cohort**

We proceeded to validate the RAD51 score as a biomarker to predict PARPi sensitivity in an independent PDX cohort (PDX cohort-2). The independent PDX panel consisted of 28 TNBC models, including eight tumors with pathogenic variants in *BRCA1* ($n = 5$), *BRCA2* ($n = 2$), or *PALB2* ($n = 1$). All these variants were classified as pathogenic by the ClinVar database (https://www.ncbi.nlm.nih.gov/clinvar/), and their variant allele frequencies (VAFs) were consistent with loss of heterozygosity (LOH; Table EV3). The antitumor activity of three PARPi (olaparib, niraparib, and/or veliparib) was analyzed in this cohort (Fig 4A). HRR capacity was quantified using the HRD score and the RAD51 score (including quantification

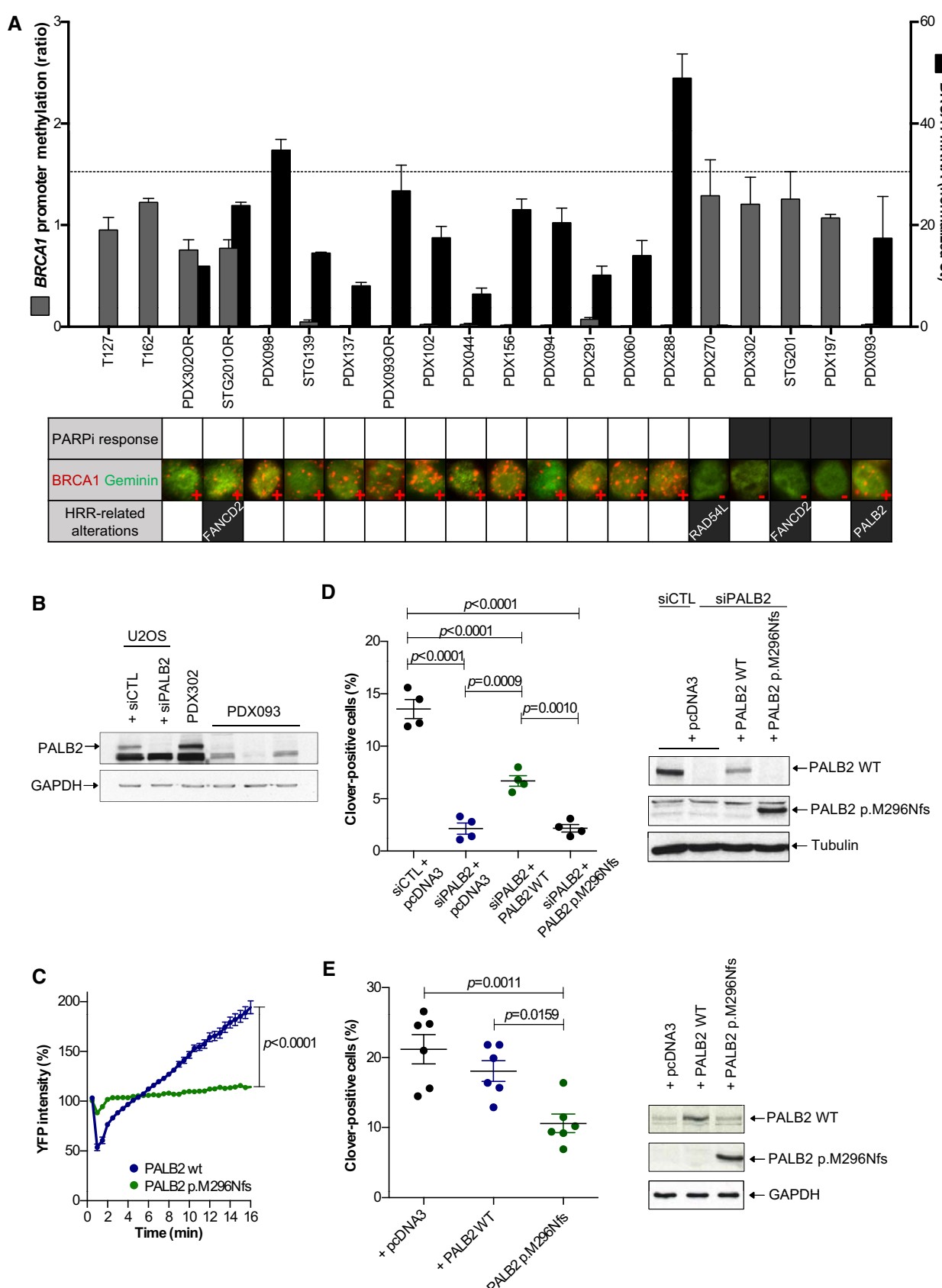

Figure 2.

**Figure 2.  HRR-related alterations in PDX cohort-1 and PARPi response.**

A    Levels of *BRCA1* promoter hypermethylation, levels of *BRCA1* mRNA, and the presence of BRCA1 nuclear foci by immunofluorescence are shown (larger views and separate channels are shown in Appendix Fig S1). T127 and T162 were used as positive controls for hypermethylated *BRCA1* promoter. Error bars indicate SEM from independent tumors ($n \geq 2$). Dashed line indicates mean of *BRCA1* mRNA levels in normal breast. PARPi response is shown in the summary underneath: white box: PD; black box: PR/CR. Alterations in HRR-related genes in PDX are also indicated.

B    Western blot of PALB2 detected in U2OS cells and PDXs. Three biological replicates of PDX093 are shown; PDX302 is used as PALB2 wild-type PDX control.

C    YFP-PALB2 recruitment to laser-induced DSBs is impaired in HeLa cells expressing PALB2 p.M296Nfs ($n = 4$, unpaired *t*-test at 16 min). Error bars indicate SEM of >40 cells per condition.

D, E    Gene targeting efficiency using Cas9/mClover-LMNA1 homologous recombination assay of (D) siRNA PALB2 cells ($n = 4$, one-way ANOVA) or (E) cells with no *PALB2* depletion complemented with wild-type and p.M296Nfs siRNA-resistant constructs ($n = 7$, one-way ANOVA). Western blots of PALB2 wild-type and PALB2 p.M296Nfs for each condition are shown. Error bars indicate SEM from independent experiments.

Source data are available online for this figure.

of RAD51 and γ-H2AX). Based on PDX cohort-1 results and previous studies (Graeser *et al*, 2010), a 10%-RAD51 score cutoff was used to differentiate PARPi-sensitive from PARPi-resistant PDXs in the PDX cohort-2. Seven models (25%) showed a RAD51 score $\leq 10\%$ and 21 models (75%) showed a RAD51 score above the cutoff, with 100% sensitivity and 100% specificity for PARPi response prediction (Fig 4A). Therefore, as in PDX cohort-1, the RAD51 score showed complete discriminative capacity in predicting PARPi response (ROC AUC = 1), while the HRD score had lower predictive power (ROC AUC = 0.735). These differences in AUC between the scores were statistically significant (difference between AUC = 0.27; Confidence Interval 95% (CI 95%) 0.08–0.46; $P = 0.005$; Fig 4B). Mechanistically, loss of 53BP1 did not explain PARPi resistance in the three *BRCA1*-mutated PDXs that exhibited RAD51 foci. Instead, two of them exhibited BRCA1 foci by immunofluorescence, indicative of potentially functional HRR restoration by BRCA1 hypomorphic variants (Fig 4C; Drost *et al*, 2016; Wang *et al*, 2016; Cruz *et al*, 2018). Altogether, these results support the use of the RAD51 assay as a predictive biomarker of PARPi response independent of the BRCA status and further demonstrate that this assay captures BRCA-related tumors that restore HRR capacity regardless of the resistance mechanism.

**Scoring RAD51 in clinical samples identifies HRR-deficient tumors among patients with hereditary breast and ovarian cancer (HBOC) syndrome, including PALB2-related tumors**

To assess the ability of the RAD51 assay to identify HRR-deficient tumors beyond gBRCA mutations, we scored for RAD51 in 23 FFPE tumor samples from a cohort of patients with clinical suspicion of hereditary breast cancer and without gBRCA mutations (Fig 5A). Six tumors derived from young patients ($\leq 35$ years), and the remaining 17 were obtained from 14 patients with family history of BC. We enriched our patient cohort with 11 tumors from patients harboring germline mutations in *PALB2* (gPALB2; Fig 5B, Table EV4). We analyzed γ-H2AX, BRCA1, and RAD51 nuclear foci in these tumors (Fig 5B and C). Fourteen out of 23 tumor samples

showed low RAD51 score ($\leq 10\%$ cutoff), including all the eleven gPALB2 tumor samples. The three tumors with low RAD51 score that lacked gPALB2 mutations showed lack of BRCA1 nuclear foci, raising the possibility of *BRCA1* promoter hypermethylation as being the cause of HRR deficiency (Pt02, Pt07, Pt11; Fig 5B and C). These data showed that carrying a *PALB2* mutation is associated with higher odds of displaying low RAD51 score [odds ratio (OR) = 62.4; CI 95% 2.852–1367; $P = 0.0003$]. Altogether, our results demonstrate that the RAD51 assay identifies HRR-deficient tumors that are sensitive to PARPi therapy beyond the gBRCA condition.

# Discussion

Our results show that RAD51 foci in PDX tumor samples correlated with PARPi response, while *BRCA1* promoter hypermethylation, *BRCA1* expression, BRCA1 foci, or HRR gene mutations did not fully correlate with PARPi response. Interestingly, the RAD51 assay was able to capture the HRR functionality in untreated tumors due to their high levels of endogenous DNA damage. A RAD51 score cutoff of 10% predicted the response to PARPi with high specificity and sensitivity, outperforming the HRD score. In clinical samples, the RAD51 assay classified as HRR-deficient all the tumors from patients with deleterious gPALB2 mutations, and three tumors from young-onset BC patients with no germline mutations in DNA repair genes. Our results support that HRR deficiency provides the basis of PARP inhibition sensitivity *in vivo* and is frequent among tumors without germline mutations in *BRCA1/2* genes, in line with the frequency of HRD genomic signature among breast cancer patients (Davies *et al*, 2017).

PARP inhibitors have become the paradigm of drug-mediated synthetic lethality in HRR-deficient tumors and have shown clinical efficacy in patients with *BRCA1/2*-related breast and ovarian cancers (Fong *et al*, 2009; Audeh *et al*, 2010; Tutt *et al*, 2010; Robson *et al*, 2017). In addition, PARPi are beneficial as maintenance treatment in ovarian cancer patients with platinum-sensitive relapse (Mirza *et al*, 2016; Coleman *et al*, 2017; Pujade-Lauraine *et al*, 2017).

**Figure 3.  Lack of RAD51 nuclear foci identifies PARPi-sensitive PDX tumors.**

A    Percentage of geminin-positive, RAD51 nuclear foci-containing cells detected by immunofluorescence in FFPE samples from PDX tumors treated with vehicle or PARPi (larger views and separate channels are shown in Appendix Fig S2). Error bars indicate SEM from independent tumors ($n \geq 2$). PARPi response is shown in the summary underneath: white box: PD; black box: PR/CR. Immunofluorescence staining of RAD51 foci in PARPi- and vehicle-treated PDX tumors is shown. Alterations in HRR-related genes are also summarized: hBRCA1: *BRCA1* promoter hypermethylation and lack of *BRCA1* expression and BRCA1 nuclear foci formation.

B    Restoration of RAD51 foci formation in PARPi acquired-resistant PDXs. Immunofluorescence staining of BRCA1 and RAD51 foci in PARPi-treated tumors from STG201, PDX302, and the corresponding PARPi acquired-resistant models (STG201OR and PDX302OR). Scale bars: 10 μm.

C    Quantification of geminin-positive cells that exhibit γ-H2AX nuclear foci following treatment with vehicle and olaparib. Paired *t*-test in PARPi-sensitive (CR/PR) versus PARPi-resistant (PD) PDXs.

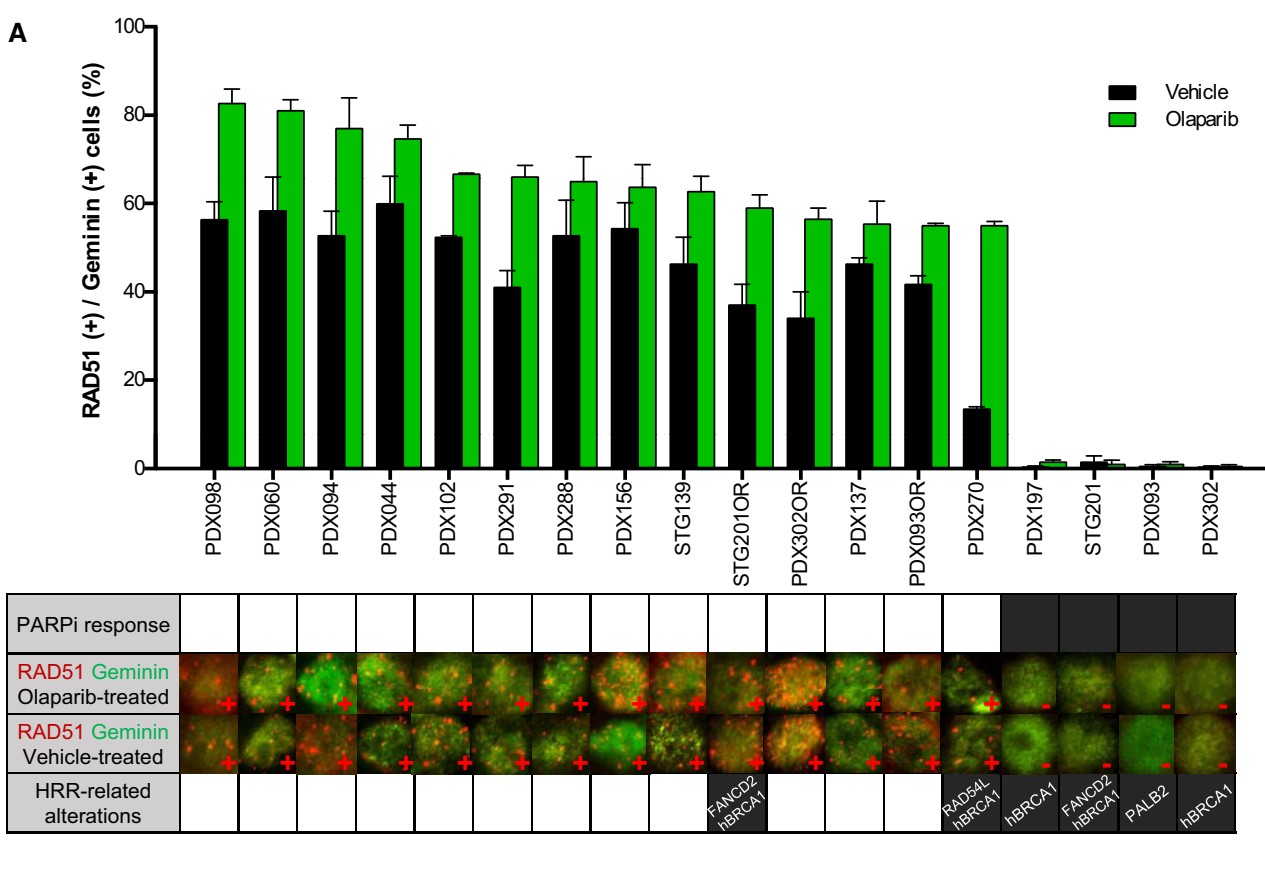

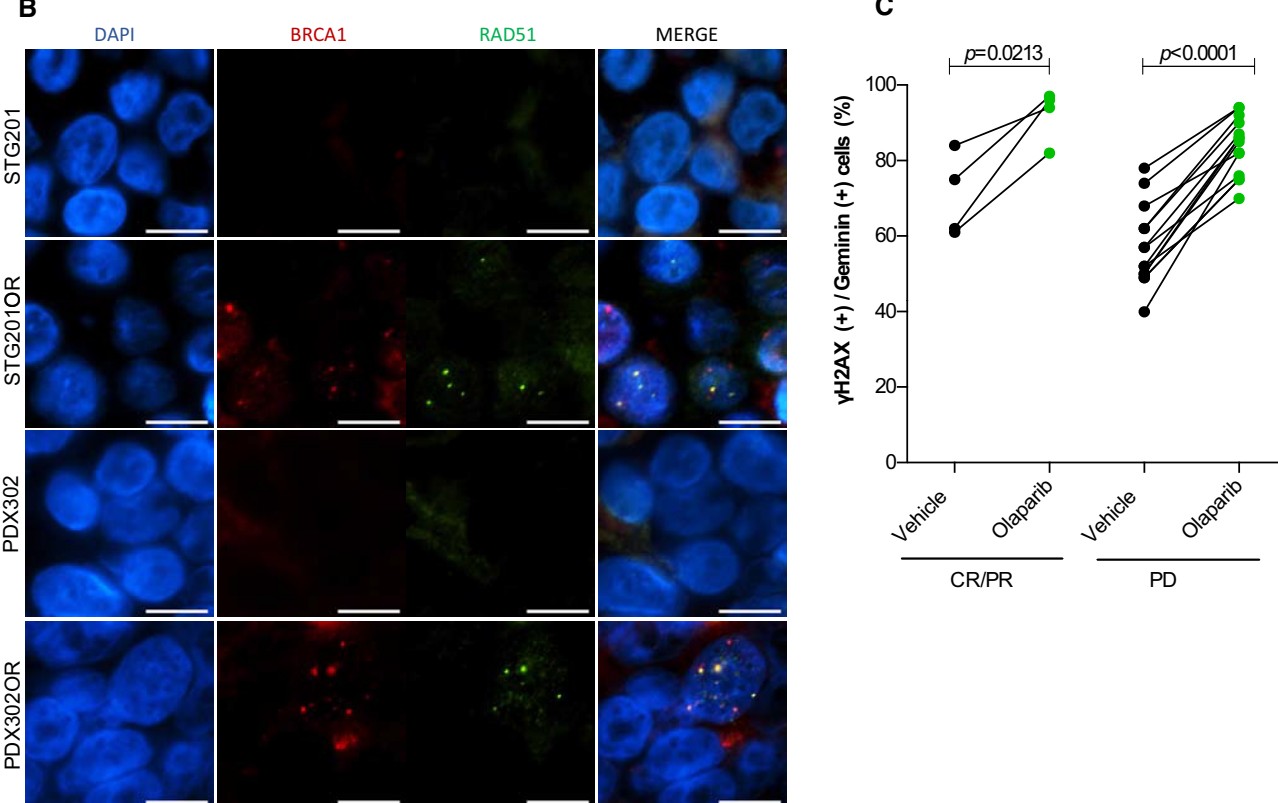

**Figure 3.**

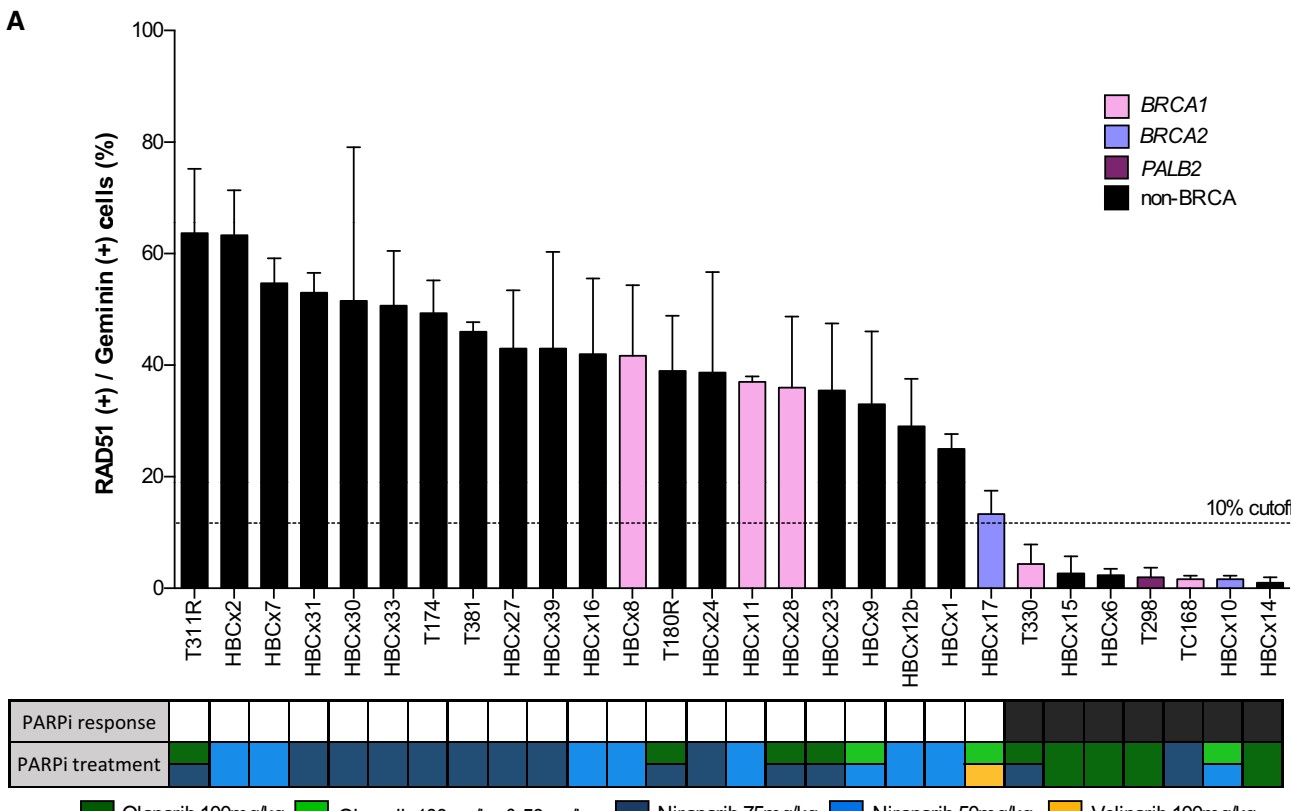

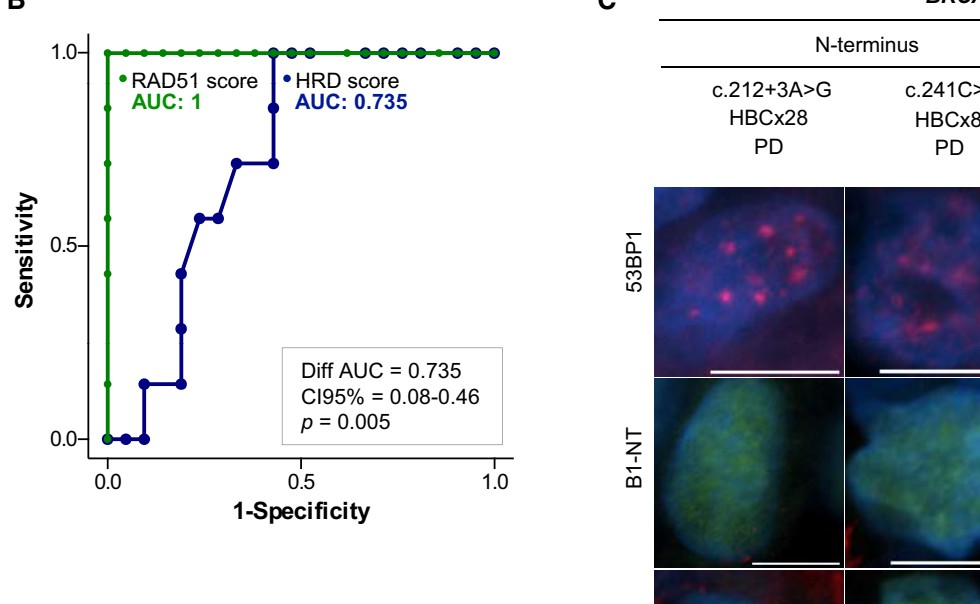

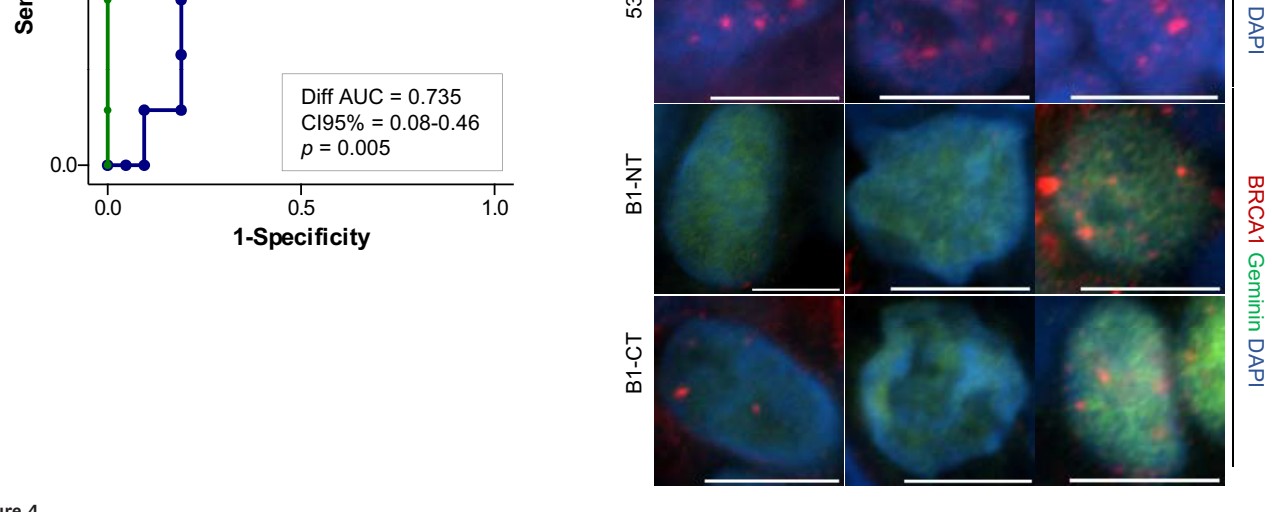

**Figure 4.**

**Figure 4.  RAD51 score predicts PDX response to PARPi in an independent PDX panel (cohort-2).**

A  Percentage of geminin-positive, RAD51 nuclear foci-containing cells in FFPE samples from untreated PDX tumors. Color bars indicate the presence of pathogenic variants in the indicated genes. Error bars indicate SEM from independent tumors ($n \geq 2$). PARPi response is shown in the summary underneath: black box: PR/CR; white box: PD. Box colors indicate the PARP inhibitor treatment. Boxes with two colors indicate the same response to both treatments. Olaparib 100 and 50 mg/kg indicates that both doses were tested and resulted in the same response categorization.

B  ROC curves of the RAD51 score and HRD score, for PARPi response prediction capacity in the PDX cohort-2. Bootstrap statistical test.

C  Immunofluorescence staining of 53BP1 and BRCA1 nuclear foci [with an antibody toward the N-terminus (B1-NT) or C-terminus (B1-CT) of BRCA1] in three BRCA1-mutant, PARPi-resistant models from PDX cohort-2. The location of the mutation within the gene is indicated. Scale bars: 10 μm.

Overall, these results have encouraged the medical community to explore the activity of PARPi beyond BRCA1/2-related malignancies toward other tumor types showing deficiency in HRR (Mateo et al, 2015). In this sense, there is a clear need to develop robust and clinically feasible biomarkers of HRR functionality correlating with treatment response. Recent advances toward the development of biomarkers of response and resistance to PARPi have been based on targeted sequencing of DNA repair genes, genomic scars, or gene and protein expression (Konstantinopoulos et al, 2015). Nevertheless, some of these biomarkers have pitfalls. For example, genomic signatures have a limited capacity to capture restoration of HRR functionality that may occur during tumor evolution or after drug pressure. Instead, functional assays of HRR status provide a more comprehensive and dynamic readout of tumor HRR capacity throughout disease evolution and at the specific moment of treatment decision (Watkins et al, 2014; Konstantinopoulos et al, 2015). RAD51 foci formation had been proposed as a predictive biomarker of PARPi response; however, the assay required tumor biopsies either collected after patient exposure to DNA damaging agents or irradiated ex vivo (Graeser et al, 2010; Naipal et al, 2014). Here, we report on the performance of the RAD51 assay in FFPE cancer samples without prior patient treatment or exogenous DNA damage induction and show its correlation with PARPi sensitivity. This functional assay provides an accurate measurement of HRR status and PARPi sensitivity at the time of treatment decision-making. Furthermore, it surpasses the common limitations of previous functional assays and facilitates its transferability from the research setting to the clinical diagnosis.

The RAD51 assay has some limitations: firstly, when PARPi sensitivity occurs via mechanisms that do not directly impact on the ability of cells to perform HRR, e.g., alterations in ATM (Chen et al, 2017; Davies et al, 2017; preprint: Balmus et al, 2018) or in the RNASEH2 complex (Zimmermann et al, 2018); secondly, when PARPi sensitivity occurs via mechanisms that preserve RAD51 foci formation, e.g., alterations in the MRN complex, RAD51AP1, polymerase eta, or ERCC1 (Kawamoto et al, 2005; Wiese et al, 2007; Oplustilova et al, 2012; Postel-Vinay et al, 2013); thirdly, when HRR-deficient tumors have acquired PARPi resistance via RAD51-independent mechanisms such as loss of PARG, mutations in PARP1, or those that involve replication fork stabilization (Guillemette et al, 2015; Chaudhuri et al, 2016; Kais et al, 2016; Yazinski et al, 2017; Gogola et al, 2018; Michelena et al, 2018; Pettitt et al, 2018); fourthly, when a tumor has low proliferation index or low endogenous DNA damage, in which cases the assay would not be feasible.

In summary, we demonstrate the feasibility of the RAD51 assay in routine FFPE tumor samples and its utility to identify several populations that might be sensitive to PARPi. First, the germline population with HRR alterations, including BRCA1/2, PALB2 and probably other genes, such as RAD51C or RAD51D. In these patients, the RAD51 assay could be used as an enrichment biomarker to better predict sensitivity to PARPi, since restoration of the HRR pathway might have occurred and result in PARPi resistance (Konstantinopoulos et al, 2015; Cruz et al, 2018). Second, tumors with somatic alterations in HRR-related genes, such as the PALB2 mutations described in 4% of metastatic BC (Lefebvre et al, 2016; Lee et al, 2018). And third, tumors with epigenetic HRR silencing, such as BRCA1 promoter hypermethylation. Our data from PDXs strongly support the clinical development of PARPi in non-gBRCA BC patients, as other studies have shown PDXs to effectively capture clinical responses (Izumchenko et al, 2017). Interestingly, there is a clinical trial recruiting patients with advanced BC with BRCA1/2 promoter hypermethylation to analyze the response to olaparib monotherapy, which includes tumor sample collection for biomarker analysis (NCT03205761). Additional work is needed to define the sensitivity and specificity of the RAD51 assay to predict PARPi benefit in the clinic and lead to better selection of patients for PARP inhibition treatment.

# Materials and Methods

### Patient-derived xenograft (PDX) models in cohort-1

Fresh tumor samples from 13 breast cancer patients without known gBRCA mutation were collected prospectively for implantation into nude mice at VHIO under an institutional review board (IRB)-approved protocol and the associated informed consent. The experiments conformed to the principles of the WMA Declaration of Helsinki and the Department of Health and Human Services Belmont Report were conducted following the European Union's animal care directive (2010/63/EU) and were approved by the Ethical Committee of Animal Experimentation of the Vall d'Hebron Research Institute. Fresh primary or metastatic human breast tumors were obtained from patients at time of surgery or biopsy and immediately implanted into the mammary fat pad (surgery samples) or the lower flank (metastatic samples) of 6-week-old female athymic HsdCpb:NMRI-Foxn1nu (Harlan Laboratories) or NOD.Cg-Prkdc$^{scid}$Il2rg$^{tm1Wjl}$/SzJ (Charles River) mice. Animals were continuously supplemented with 1 μM 17β-estradiol (Sigma-Aldrich) in their drinking water. Upon growth of the engrafted tumors, the model was perpetuated by serial transplantation onto the lower flank. In each passage, flash-frozen and formalin-fixed paraffin-embedded (FFPE) samples were taken for genotyping and histological studies. Two models (1 TNBC, STG139 and 1 ER-positive BC, STG201) were generated in CRUK/UCAM, a member of the EurOPDX consortium (http://www.europdx.eu), as previously reported (Bruna et al, 2016). Three models with acquired PARPi-resistance were generated in the laboratory (see below).

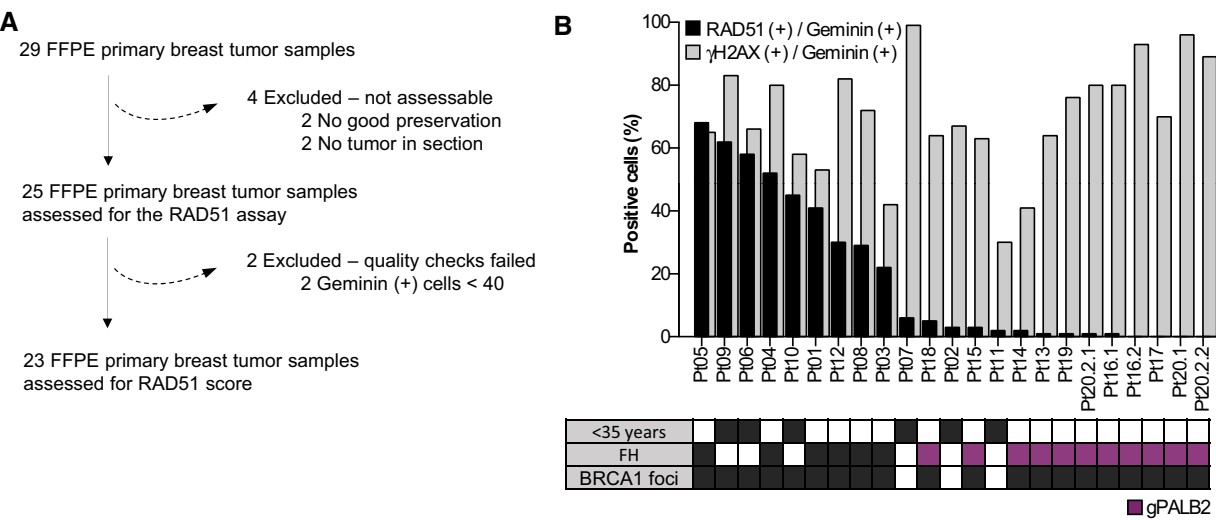

**Figure 5.**

Figure 5.  RAD51 score in tumors from patients with HBOC syndrome, including germline *PALB2* mutation carriers.
A   Consort diagram of the collected and analyzed FFPE tumor samples from patients with HBOC syndrome following the scoring criteria for the RAD51 assay.
B   Percentage of geminin-positive, RAD51, or γ-H2AX nuclear foci-containing cells in FFPE tumor samples from patients with HBOC syndrome. The box underneath summarizes the patient's young onset (< 35 years), her family history (FH, purple box for PALB2-related tumors) and the presence of BRCA1 nuclear foci in the analyzed tumor samples.
C   Immunofluorescence staining of γ-H2AX, BRCA1, and RAD51 foci in three representative FFPE tumors from patients: one RAD51- and BRCA1-positive tumor (Pt03), one RAD51- and BRCA1-negative tumor (Pt11), and one PALB2-related tumor, with BRCA1 but not RAD51 nuclear foci (Pt20.1). Scale bars: 10 μm.

### *In vivo* experiments of PDX sensitivity to PARPi

To evaluate the sensitivity to PARP inhibition, at least three tumor-bearing mice were equally distributed into treatment groups with tumors ranging 50–350 mm$^3$. When allocating animals to treatment arms, we ensured that the mean starting volume between arm was not statistically different by *t*-test (see Appendix Table S1). Olaparib 50 mg/kg oral (p.o.) was administered 6 days per week (in 10% v/v DMSO/10% w/v Kleptose [HP-β-CD]; ter Brugge *et al*, 2016). To generate PDX models with acquired resistance to PARPi, olaparib treatment was maintained in olaparib-sensitive tumors until individual tumors regrew. Tumor growth was measured blinded to the treatment effect with caliper bi-weekly from first day of treatment to day 21 and every 7–10 days in the acquired resistance setting. Mouse weight was recorded twice weekly. The tumor volume was calculated as $V = 4\pi/3/L\times l\times l$, "L" being the largest diameter and "l" the smallest. Mice were euthanized when tumors reached 1,500 mm$^3$, in accordance with institutional guidelines. The antitumor activity was determined by comparing tumor volume at 21 days to its baseline: % tumor volume change = $(V_{day21}-V_{day1})/V_{day1} \times 100$. For olaparib-sensitive PDXs, the best response was defined as the minimum value of % tumor volume change sustained for at least 10 days. To classify the antitumor response, the Response Evaluation Criteria in Solid Tumors (RECIST) on the % tumor volume change was modified and labeled as mRECIST: CR (complete response), best response ≤ −95%; PR (partial response), −95% < best response ≤ −30%; SD (stable disease), −30% <best response ≤+20%, PD (progressive disease), % tumor volume change at day 21 >+20% (Therasse *et al*, 2000; Gao *et al*, 2015).

### PDX cohort-2 for RAD51 assay validation

To validate the RAD51 assay, we used an independent PDX cohort whose response to PARPi was tested at XenTech. These PDXs were generated at Curie Institute (Paris, France) and Paoli Calmette Institute (Marseille, France) under approved informed consent. The majority of these PDXs were previously published (Marangoni *et al*, 2007; Charafe-Jauffret *et al*, 2013). For *in vivo* experiments, when tumors reached a size of 70–250 mm$^3$, mice were randomly assigned to homogeneous groups of 5–10 animals and were treated p.o. with niraparib (50 or 75 mg/kg), olaparib (50 or 100 mg/kg), or veliparib (100 mg/kg) daily for 28 days. Tumor volume was evaluated by measuring bi-weekly tumor diameters with a caliper.

### HRD score

Myriad's myChoice® HRD test was performed at Myriad Genetics on DNA extracted from PDXs of cohort-2. DNA extraction was performed by using the NucleoBond AXG100 kit (Macherey-Nagel).

### Exome sequencing

All laboratory methods were performed using the manufacturer's protocols. Genomic DNA was isolated from fresh-frozen PDX tissue using the Promega Maxwell 16® Tissue SEV DNA Purification Kit (catalog #AS1030) and the Maxwell® 16 MDx Instrument (Promega Corp., Madison, WI, USA). Specifically, samples were loaded into well #1 of the Maxwell cartridge, run using the "DNA/tissue" protocol, and genomic DNA was eluted with 300 μl Elution Buffer. All samples were quantified using the Qubit® dsDNA HS Assay Kit (catalog #Q32851) and Qubit 2.0 fluorometer (Thermo Fisher Scientific, Waltham, MA, USA). Exome libraries were constructed using the KAPA Hyper Prep Library Preparation Kit (Kapa Biosystems Inc., Wilmington, MA, USA), and genes were captured using the xGen® Exome Research Panel v1.0 (Integrated DNA Technologies, Coralville, IA, USA). Paired-end 150 bp sequencing was performed on an Illumina HiSeq 4000 using TruSeq SBS reagents (Illumina) with approximately 10 Gbp per sample for ~200-fold average sequence depth.

Data analysis followed standard methodologies. Briefly, sequencing reads were aligned to both human hg19 and mouse mm10 genomes using Burrows-Wheeler Alignment (BWA), and then, mouse-derived sequences in the human.bam file were removed using Disambiguate (Ahdesmäki *et al*, 2016). Variants were called in the human.bam files using VarDirect (Lai *et al*, 2016). Copy number analysis was performed using Seq2C (Reznik *et al*, 2016).

### 73-gene profiling of XenTech's PDX samples

Mutation profiling of 73 genes among the most frequently mutated in cancer according to the COSMIC database was performed at BGI (Beijing, China) on genomic DNA by exon trapping with NimbleGen microarray followed by deep sequencing by using Illumina's HiSeq technology, with at least 50× effective mean depth for each sample.

### *BRCA1* promoter methylation

*BRCA1* promoter methylation was measured using methylation-specific multiplex ligation-dependent probe amplification (MS-MLPA; MRC Holland, Amsterdam, the Netherlands) according to manufacturer's instructions. The two xenografts generated in CRUK/UCAM (STG139 and STG201) had been previously tested using reduced-representation bisulfite sequencing (RRBS; Bruna *et al*, 2016) and further validated using MS-MLPA. Positive controls

of *BRCA1* promoter hypermethylated were used (T127 and/162; ter Brugge *et al*, 2016).

### *BRCA1* mRNA expression

RNA was extracted from PDX samples (15–30 mg) by using the PerfectPure RNA Tissue kit (five Prime). The purity and integrity were assessed by the Agilent 2100 Bioanalyzer system, and cDNA was obtained using the PrimeScript RT Reagent kit (Takara). Quantitative RT–PCR was performed in a 7900HT Fast Real-Time PCR System (Applied Biosystems) using TaqMan Universal Master Mix II (Applied Biosystems) and predesigned human specific primers and TaqMan probes (Hs99999908_m1 for *GUSB*, Hs99999903_m1 for *ACTB*, and Hs01556193_m1 for *BRCA1*). The comparative CT method was used for data analysis, in which geNorm algorithms were applied to select the most stably expressed housekeeping genes (*GUSB* and *ACTB*) and geometric means were calculated to obtain normalized CT values (Vandesompele *et al*, 2002).

### Immunofluorescence

The following primary antibodies were used for immunofluorescence: rabbit anti-RAD51 (Santa Cruz Biotechnology sc-8349 1:250), rabbit anti-RAD51 (Abcam ab133534, 1:1000), mouse anti-geminin (NovoCastra NCL-L, 1:100 in PDX samples, 1:60 in patient samples), rabbit anti-geminin (ProteinTech 10802-1-AP, 1:400), mouse anti-BRCA1 (Santa Cruz Biotechnology sc-6954, 1:50), mouse anti-BRCA1 (Abcam ab16780, 1:200), mouse anti-γ-H2AX (Millipore #05-636, 1:200), rabbit anti-53BP1 (Cell Signalling #4937, 1:100). Goat anti-rabbit Alexa fluor 568 (Invitrogen; 1:500), goat anti-mouse Alexa fluor 488 (Invitrogen; 1:500), donkey anti-mouse Alexa fluor 568 (Invitrogen; 1:500), and goat anti-rabbit Alexa fluor 488 (Invitrogen; 1:500) were used as secondary antibodies. For target antigen retrieval, sections were microwaved for 4 min at 110°C in DAKO Antigen Retrieval Buffer pH 9.0 in a T/T MEGA multifunctional Microwave Histoprocessor (Milestone). Sections were cooled down in distilled water for 5 min, then permeabilized with DAKO Wash Buffer (contains Tween-20) for 5 min, followed by incubation in blocking buffer (DAKO Wash Buffer with 1% bovine serum albumin) for 5 min. Primary antibodies were diluted in DAKO Antibody Diluent and incubated at room temperature for 1 h. Sections were washed for 5 min in DAKO Wash Buffer followed by 5 min in blocking buffer. Secondary antibodies were diluted in blocking buffer and incubated for 30 min at room temperature. The 2-step washing was repeated followed by 5-min incubation in distilled water. Dehydration was performed with increasing concentrations of ethanol. Sections were mounted with DAPI ProLong Gold antifading reagent and stored at −20°C. Immunofluorescence images were acquired using Olympus DP72 microscope and generated using CellSens Entry software.

RAD51 foci of 0.42–1.15 μm diameter were quantified on FFPE PDX or patient tumor samples, by scoring the percentage of geminin-positive cells with 5 or more RAD51 nuclear foci. Geminin is a master regulator of cell-cycle progression that ensures the timely onset of DNA replication and prevents its rereplication, and used as counterstaining to mark for S/G2-cell cycle phase (Ballabeni *et al*, 2013). Scoring was performed blindly onto life images using a 60×-immersion oil lens. One hundred geminin-positive cells from

at least three representative areas of each sample were analyzed. At least two biological replicates of each PDX model (both vehicle- and olaparib-treated) were analyzed. The amount of DNA damage was quantified on FFPE PDX tumor samples by scoring the percentage of geminin-positive cells with γ-H2AX foci, as described for RAD51 scoring. Samples with low γ-H2AX (< 25% of positive cells) or with < 40 geminin-positive cells were not evaluated (Fig EV1B).

### Cell lines

U2OS osteosarcoma cells (HTB-96) were purchased from American Type Culture Collection (ATCC) and maintained in McCoy's 5A (Gibco) supplemented with 10% Fetal Bovine Serum (FBS) and 1% penicillin/streptomycin (P/S). HeLa cells were maintained in DMEM medium (Corning) supplemented with 10% FBS and 1% P/S. All cell lines were routinely tested to be mycoplasma free.

### Cas9/mClover-LMNA homologous recombination assay

The mClover-LMNA homologous recombination assay was adapted from Pinder *et al* (2015) and Pauty *et al* (2017). In brief, U2OS cells were seeded at 175,000 cells per well in 6-well plates to be transfected 6 h later with control or *PALB2* siRNA at a final concentration of 50 nM using Lipofectamine RNAiMAX (Invitrogen). Twenty-four hours post-transfection, $1 \times 10^6$ cells per condition were pelleted and resuspended in 100 μl complete nucleofector solution (SE Cell Line 4D-Nucleofector™ X Kit, Lonza) to which 1 μg of pCR2.1-CloverLMNAdonor, 1 μg pX330-LMNAgRNA, 1 μg of the indicated *PALB2* construct, 0.1 μg of piRFP670-N1 (used as transfection control), and 200 ρmol of siRNA were added. Once transferred to a 100 μl Lonza certified cuvette, cells were transfected using the 4D-Nucleofector X-unit, program CM-104, immediately resuspended in culture media, and transferred to a 100 mm dish for 64 h. Then, 500,000 cells were plated onto glass coverslips, while the remaining was lysed for Western blotting as described below. Coverslips were fixed with 4% paraformaldehyde and analyzed for Clover expression by fluorescence microscopy a total of 72 h post-nucleofection.

### Protein extraction and Western blotting

U2OS cells were resuspended in ice-cold lysis buffer (50 mM Tris–HCl, pH 7.4, 500 mM NaCl, 0.5% NP-40) containing protease and phosphatase inhibitors (PMSF (1 mM), aprotinin (4 μg/ml), leupeptin (1 μg/ml), NaF (5 mM), and $Na_3VO_4$ (1 mM)). Frozen tumors of each PDX ($n = 1$–3 replicates) were lysed in 500 μl of ice-cold lysis buffer (40 mM Tris–HCl pH 7.4, 1% Triton X-100, 40 mM Beta-glycero phosphate, 5% Glycerol, 100 mM NaCl, 1 mM EDTA, 50 mM NaF, and protease and phosphatase inhibitors as above) per 100 mg and then crushed using a sterile pestle (Axygen). U2OS and PDX lysates were incubated for 30 min on ice and sonicated 30 s ON\OFF for 10 cycles with a Bioruptor (Diagenode). Insoluble material was removed by high-speed centrifugation at 4°C, and protein concentration was determined by the Bradford assay. Total soluble protein extracts were separated by SDS–PAGE and transferred onto nitrocellulose

membranes (GE Healthcare). Membranes were blocked for an hour at room temperature with 5% non-fat dry milk in PBST and probed overnight, 4°C, with rabbit polyclonal PALB2 antibody (Bethyl, A301-246A) at 1:2,000 and mouse monoclonal GAPDH antibody (Fitzgerald, 10R-G109a) at 1:160,000 in the blocking solution. Horseradish peroxidase-conjugated secondary antibodies (Jackson ImmunoResearch) were used at 1:10000 in PBST for 1 h at room temperature followed by detection using the Western Lighting Chemiluminescence Reagent Plus (PerkinElmer).

### Localization of PALB2 to laser-induced DSBs

The experiments were performed as described in Couturier *et al* (2016). Briefly, HeLa cells were transfected with YFP-PALB2-WT or YFP-PALB2-p.M296Nfs and microirradiated. The recruitment of YFP-PALB2 to laser-induced DNA damage sites was monitored over time.

### HBOC patient cohort

The cohort consisted of breast cancer patients from the Vall d'Hebron University Hospital, with FFPE material representative of the disease and signed IRB-approved informed consent form. Due to personal or family history, and after ruling out *BRCA1/2* mutations, patients were tested for germline mutations linked to breast cancer susceptibility within a research protocol. Immunofluorescence analysis and RAD51 quantification was performed as described for FFPE PDX tumor samples.

### Statistical analysis

Regarding the sample size calculation, this exploratory study involved as many samples as possible during the study timeframe. Data were analyzed with GraphPad Prism version 7.0. Error bars represent the Standard Error of the Mean (SEM) of at least two biological replicates, unless otherwise stated. Shapiro–Wilk test was used to assess normality of data distributions. Statistical tests were performed using paired or unpaired two-tailed *t*-test (for two groups comparison of YFP intensity (%) in the FRAP assay and of γ-H2AX/geminin- and geminin-positive cells in PDX cohort-1), Mann–Whitney *U*-test (for two groups comparison of RAD51/geminin-positive cells in PDX cohort-1), or one-way ANOVA (for three or more groups in gene targeting efficiency comparisons using Cas9/mClover—LMNA1 homologous recombination assay). Pearson correlation was used for analyze the correlation between the RAD51 score and the tumor volume change upon olaparib treatment. For the validation of the two anti-RAD51 antibodies, Spearman correlation was used. The ROC AUC was calculated to estimate the prediction capacity of HRD and RAD51 scores to PARPi response. For ROC curve comparison, a two-sided bootstrap test was used by means of statistical package pROC in R software version 3.4.1. To calculate the association between *PALB2* mutation and RAD51 score, a logistic regression model was fitted to estimate the odds ratio (OR) with CI 95%. Levene's test is used to test whether groups of comparison have equal variances (homoscedasticity). No evidence has been found to reject the hypothesis of homoscedasticity in our data. Consequently, the statistical comparison has been carried out under the hypothesis of similar variance between groups.

## Data availability

The datasets produced in this study are available in the following database:

Exome sequencing cohort 1: Sequence data has been deposited at the European Genome-phenome Archive (EGA), which is hosted by the EBI and the CRG, under accession number EGAS00001003267 (https://ega-archive.org/studies/EGAS00001003267).

73-gene profiling cohort 2: Sequence data has been deposited at the European Nucleotide Archive (ENA), under accesion number PRJEB28816 (http://www.ebi.ac.uk/ena/data/view/PRJEB28816).

**Expanded View** for this article is available online.

## Acknowledgements

The authors thank Dr. Geoffrey Shapiro, Dr. Peter Bouwman, Dr. Neil Johnson, Dr. Josep V. Forment, the Molecular Pathology Group at VHIO, the Breast Cancer and Melanoma Group at VHIO, Dr. Felip Vilardell Villellas, and Laura Duran-Lozano for helpful discussions. The Breast Surgical Unit from Vall d'Hebron Hospital; Pilar Antón, Maite Calvo, Patricia Cozar from the Experimental Therapeutics Group at VHIO; Brian Dougherty, Zhongwu Lai, and Ambar Ahmed from AstraZeneca and Amélie Rodrigue and Yan Coulombe from Masson laboratory have provided technical support. The authors acknowledge the Cellex Foundation for providing research facilities and equipment. This Research Project was also supported by ESMO with the aid of a grant from Roche. Any views, opinions, findings, conclusions, or recommendations expressed in this material are those solely of the authors and do not necessarily reflect those of ESMO or Roche. This research was supported by Spanish Instituto de Salud Carlos III (ISCIII) funding, an initiative of the Spanish Ministry of Economy and Innovation partially supported by European Regional Development FEDER Funds [FIS PI17/01080 to V. Serra, FIS PI12-02606 to J. Balmaña, FIS PI12/02585 and PI15-00355 to O. Diez, FIS PI13/01711 and PI16/01218 to S. Gutiérrez-Enríquez], by a TRANSCAN-2 [AC15/00063 to V.Serra], the Asociación Española Contra el Cancer (AECC) [LABAE16020PORTT to V.Serra], the Catalan Agency AGAUR [2017 SGR 540 coordinator V. Serra], the Canadian Institutes of Health Research (CIHR) [Foundation grant to J.-Y. Masson], and the Programme de soutien à des initiatives internationales de recherche et d'innovation (SIIRI) du Ministère de l'Économie, de l'Innovation et des Exportations du Québec [to J.-Y. Masson and J. Simard, PSR-SIIRI-949]. We acknowledge the "Tumor Biomarkers Collaboration" supported by the Banco Bilbao Vizcaya Argentaria (BBVA) Foundation, the GHD-Pink program, the FERO Foundation, and the Orozco Family for supporting this study [to V. Serra and J. Baselga]. M. Castroviejo-Bermejo is awarded with a Junta Provincial de Barcelona, Asociación Española Contra el Cáncer (AECC) fellowship. C. Cruz [AIOC15152806CRUZ] and S. Bonache are recipients of AECC fellowships. A. Llop-Guevara is awarded with a PERIS fellowship from the Departament de Salut, Generalitat de Catalunya [SLT002/16/00477]. V. Serra [CP14/00228] and S. Gutiérrez-Enríquez [CP10/00617] are supported by the Miguel Servet Program (ISCIII). J.-Y. Masson is a FRQS chair in genome stability, and J. Simard is a Canada Research Chair in Oncogenetics. The xenograft program in the Caldas laboratory is supported by Cancer Research UK and also received funding from an EU H2020 Network of Excellence (EuroCAN).

## The paper explained

### Problem

PARP inhibitors (PARPi) have become the paradigm of targeted drug-mediated synthetic lethality in tumors with homologous recombination repair (HRR) deficiency, showing clinical efficacy in patients with BRCA1/2-related breast and ovarian cancers, or as maintenance treatment in ovarian cancer patients with platinum-sensitive relapse. Nonetheless, non-BRCA1/2-related breast tumors may also harbor HRR deficiency and exhibit PARPi sensitivity. Since the underlying causes of HRR deficiency might be diverse, a robust biomarker to identify those tumors with impaired HRR would solve the unmet clinical need and support the expansion of PARPi use beyond the germline BRCA1/2 (gBRCA) condition. This HRR functional assay needs to be feasible in routine clinical samples and able to capture dynamic changes in HRR to accurately guide therapeutic decisions.

### Results

Using two independent breast cancer patient-derived tumor xenograft (PDX) panels, we assessed the antitumor activity of three PARPi and showed the potential of an immunostaining assay of the HRR protein RAD51 to predict PARPi sensitivity. Remarkably, all sensitive models that developed acquired PARPi resistance restored the capacity to form RAD51 foci, supporting that this assay is a dynamic biomarker of HRR functionality. In addition, we showed that the RAD51 score was superior predicting PARPi response than other genetic or genomic tests (i.e., the presence of HRR genetic alterations or the homologous recombination deficiency (HRD) score). Finally, we demonstrated that the RAD51 assay is feasible in formalin-fixed paraffin-embedded (FFPE) routine breast cancer samples and accurately classified as HRR-deficient all the PALB2-related tumors.

### Impact

The medical community is encouraged to explore the activity of PARPi beyond BRCA1/2-related malignancies toward other tumor types showing deficiency in HRR, but the currently available biomarkers to select patients have a limited positive predictive value. Our data, showing that the RAD51 assay has high sensitivity and specificity, are therefore of primary translational relevance. The RAD51 functional assay provides an accurate measurement of HRR status "in real time" and surpasses the common problems of previous functional assays, i.e., by using a method based on routine FFPE tumor samples that does not require prior DNA damaging treatment to assess the presence of RAD51 foci in the nucleus. This is crucial in order to design the transferability of this assay from the research setting to the clinical diagnosis. Overall, this work unveils the RAD51 assay as a functional biomarker to improve patient selection for PARPi monotherapy in other cancers and beyond the gBRCA condition, including those with *PALB2* mutation.

## Author contributions

MC-B, CCr, MJO, JBl, and VS designed the study. MC-B, CCr, AL-G, MD, YHI, AG-O, BP, MG, OR, JG, SB, and AM-F performed experiments or procedures. MC-B, CCr, AL-G, SG-E, and AB analyzed the data. GV, CV, RD, CCr, and SG-E provided and analyzed patients' data. PG, MV, VP, XS-C, GD, JS, and OD provided study materials. PN, ITR, JCB, CCa, CS, JC, JJ, J-YM, SC, J-GJ, MJO, OD, JBl, and VS supervised experiments. MC-B, CC, JBl, and VS wrote the manuscript, with input and scientific advice from AL-G, SG-E, CCa, JBas, J-YM, SC, MJO, and OD.

## Conflict of interest

V.S. declares a non-commercial research agreement with AstraZeneca and Tesaro. J.B.l. is an advisory board member for Clovis, Tesaro, and Medivation and has received speaker bureau honoraria by AstraZeneca. J.C.B. and M.J.O. are employees of AstraZeneca. S.C. and J.-G.J. are employees of XenTech. CCa is on the External Science Panel of AstraZeneca. JBas is on the Board of Directors of Foghorn and is a past board member of Varian Medical Systems, Bristol-Myers Squibb, Grail, Aura Biosciences and Infinity Pharmaceuticals. He has performed consulting and/or advisory work for Grail, PMV Pharma, ApoGen, Juno, Lilly, Seragon, Novartis and Northern Biologics. He has stock or other ownership interests in PMV Pharma, Grail, Juno, Varian, Foghorn, Aura, Infinity, ApoGen, as well as Tango and Venthera, for which is a co-founder. He has previously received Honoraria or Travel Expenses from Roche, Novartis, and Lilly. No potential conflict of interests were disclosed by the other authors.

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

### List of affiliations

1   Experimental Therapeutics Group, Vall d'Hebron Institute of Oncology, Barcelona, Spain
2   High Risk and Familial Cancer Group, Vall d'Hebron Institute of Oncology, Barcelona, Spain
3   Department of Medical Oncology, Hospital Vall d'Hebron, Universitat Autònoma de Barcelona, Barcelona, Spain
4   Oncogenetics Group, Vall d'Hebron Institute of Oncology, Barcelona, Spain
5   Genome Stability Laboratory, CHU de Québec Research Center, Québec City, QC, Canada
6   Department of Molecular Biology, Medical Biochemistry and Pathology, Laval University Cancer Research Center, Québec City, QC, Canada
7   CHU de Quebec - Université Laval Research Center, Genomics Center, CHUL, Québec City, QC, Canada
8   Department of Medical Oncology, University Hospital of Parma, Parma, Italy
9   Cancer Research UK Cambridge Institute and Department of Oncology, Li Ka Shing Centre, University of Cambridge, Cambridge, UK
10  Oncology Data Science (OdysSey Group), Vall d'Hebron Institute of Oncology, Barcelona, Spain
11  Breast Cancer and Melanoma Group, Vall d'Hebron Institute of Oncology, Barcelona, Spain
12  Pathology Department, Vall d'Hebron University Hospital, Barcelona, Spain
13  CIBERONC, Instituto de Salud Carlos III, Madrid, Spain
14  Department of Radiology, Hospital Vall d'Hebron, Universitat Autònoma de Barcelona, Barcelona, Spain
15  Department of Pathology, Dalhousie University, Halifax, NS, Canada
16  Molecular Oncology Group, Vall d'Hebron Institute of Oncology, Barcelona, Spain
17  Breast Surgical Unit, Breast Cancer Center, Hospital Vall d'Hebron, Universitat Autònoma de Barcelona, Barcelona, Spain
18  AstraZeneca, Waltham, MA, USA
19  Breast Cancer Programme, Cancer Research UK (CRUK) Cambridge Cancer Centre, Cambridge, UK
20  Human Oncology and Pathogenesis Program (HOPP), Memorial Sloan Kettering Cancer Center, New York, NY, USA
21  Department of Medicine, Memorial Sloan Kettering Cancer Center, New York, NY, USA
22  Department of Oncology, Ramón y Cajal University Hospital, Madrid, Spain
23  Vall d'Hebron Institute of Oncology, Barcelona, Spain
24  XenTech, Evry, France
25  Division of Molecular Pathology and Cancer Genomics, The Netherlands Cancer Institute, Amsterdam, The Netherlands
26  Oncology Innovative Medicines and Early Clinical Development Biotech Unit, AstraZeneca, Cambridge, UK
27  Clinical and Molecular Genetics Area, Hospital Vall d'Hebron, Universitat Autònoma de Barcelona, Barcelona, Spain

