## [Review Process File · EMBO Molecular Medicine]

A RAD51 assay feasible in routine tumor samples calls PARP inhibitor response beyond BRCA mutation

M. Castroviejo-Bermejo, C. Cruz, A. Llop-Guevara, S. Gutiérrez-Enríquez, M. Ducy, Y. H. Ibrahim, A. Gris-Oliver, B. Pellegrino, A. Bruna, M. Guzman, O. Rodriguez, J. Grueso, S. Bonache, A. Moles-Fernández, G. Villacampa, C. Viaplana, P. Gómez, M. Vidal, V. Peg, X. Serres-Créixams, G. Dellaire, J. Simard, P. Nuciforo, I. T. Rubio, R. Dientsmann, J. C. Barrett, C. Caldas, J. Baselga, C. Saura, J. Cortés, O. Déas, J. Jonkers, J.-Y. Masson, S. Cairo, J.-G. Judde, M. J. O'Connor, O. Díez, J. Balmaña, V. Serra

Review timeline:

Submission date:	04 April 2018
Editorial Decision:	11 May 2018
Revision received:	09 August 2018
Editorial Decision:	03 September 2018
Revision received:	19 September 2018
Accepted:	25 September 2018

Editor: Céline Carret

Transaction Report:

1st Editorial Decision

11 May 2018

Thank you for the submission of your manuscript to EMBO Molecular Medicine. We have now heard back from the three referees whom we asked to evaluate your manuscript.

You will see from the comments pasted below that overall, the referees find the data to be of interest but they also request additional details and information, better data presentation as well as statistics, detailed rationale, and referee 1 points to discontinued RA51 antibody and Myriad score, suggesting that reproducibility should be ascertained.

Upon our cross-commenting exercise, referees 1 and 2 agreed that important technical and some conceptual questions were raised from the three referees that need to be addressed in a revised version of the article. Lots of work is indeed needed to make the paper up to the level we expecting EMBO Molecular Medicine and I hope that you will be keen to perform the requested revision that would, in purview, greatly strengthen the study and its impact.

We would welcome the submission of a revised version within three months for further consideration and would like to encourage you to address all the criticisms raised as suggested to improve conclusiveness and clarity. Please note that EMBO Molecular Medicine strongly supports a single round of revision and that, as acceptance or rejection of the manuscript will depend on another round of review, your responses should be as complete as possible.

I look forward to receiving your revised manuscript.

***** Reviewer's comments *****

Referee #1 (Comments on Novelty/Model System for Author):

There are some technical concerns noted in my remarks below, but the relevance of a predictive assay, if validated, would be fairly high as this an area of unmet need in the field.

Referee #1 (Remarks for Author):

The manuscript is seeking to address the need for a better predictive biomarker of PARP response than somatic or germline BRCA1/2 mutations and/or mutational signatures of HR deficiency. RAD51 IHC foci are compared with the Myriad Genetics HRD score, using PDX tumours tested for PARPi sensitivity.

This an important area of unmet need - better predictive markers of PARP and other HR deficiency exploiting therapies is much needed.

The current manuscript has some problems and falls short in a few areas, weakening the conclusions that can be drawn.

1. Repeatability. The Santa Cruz antibody used here for RAD51, has been discontinued by Santa Cruz, this reviewer believes as part of the Santa Cruz goat shutdown in 2016. This is not the fault of the authors, but it raises the question of repeatability of the present study by others. Perhaps more pertinently it would imply a predictive marker would in any case need to validate other antibodies. One possible remedy would be to re-stain the relevant materials with a currently available RAD51 antibody, to demonstrate reproducibility.

2. The Myriad HRD score is also being discontinued, leading to similar issues over the ability to reproduce or extend the current study.

3. The observations of sensitivity are claimed on the basis of absent RAD51 foci. This is a negative signal and thus carries the risk of technical issues leading to a falsely negative signal and over-estimation of PARP sensitivity. One could pose the result as a positive inverse - ie presence of RAD51 foci predicts resistance. However using the absence of signal for prediction leading to an active plan to treat, may be unreliable.

4. A more robust comparison to alternative HR detection methods, could include genome wide sequence derived signatures, such as those of HRdetect (Nik-Zainal), which may be more sensitive. The sole reliance on HRD may be a falsely flattering comparison.

Referee #2 (Remarks for Author):

This manuscript compares different approaches with which to identify tumors with functional homologous recombination repair (HRR) defects. This is an important translational research topic, since such tumors are likely to respond to therapy with PARP inhibitors. Moreover, there is clinical evidence these inhibitors have efficacy against tumors beyond those with BRCA1/2 or PALB2 mutations. The design of the study is generally well thought out and the studies well performed. The authors analyzed an initial series of breast tumor PDX models for sensitivity to PARP inhibitors, DNA signatures for HRR defects, genetic aberrations affecting some HRR related genes, and the ability to correlate sensitivity with these or with BRCA1, H2Ax, or RAD51 nuclear foci. The

evidence indicated a strong correlation with RAD51 foci so the authors then applied their findings to a test set of PDX tumors and finally to histological sections from patient tumors. Their conclusion is that a low number of otherwise untreated Geminin positive tumor cells with more than 5 RAD51 foci is the best biomarker for HRR deficient tumors/responsive to PARP inhibitors, an assay that can be performed on formalin fixed and paraffin embedded tissue. Unfortunately, the presentation of data needs to be significantly improved particularly with respect to the nuclear staining shown in order to allow meaningful conclusions to be drawn by the reader. Specific points that need to be addressed are as follows:

1. While Fig. 1 shows obvious inhibitory effects of the PARP inhibitor on growth in vivo of 4 tumors, there was at least one other tumor, which grew poorly in response to the inhibitor (Fig. 1B) Was this tumor just a slow grower or did the inhibitor have a significant inhibitory effect. If so, does it also have a functional HRR deficiency as well?
2. In Fig.2A, BRCA foci are very difficult to evaluate at the magnification shown. This problem would be worse, if the figure were reduced in size. The results in Fig.2C, D, and E need more detailed explanations since not all readers may be experienced with the assays used in these panels. In Figs. 2D and E, it would be helpful to show the relative expression levels of the exogenous proteins.
3. Fig. 3 is very important to the authors' presentation, and the results in the table appear clear. However, it is impossible at the magnification provided for the reader to discern the RAD51 foci in Fig. 3A and also difficult to discern in the panels in which the nuclei are shown at higher magnification (Fig.3B). There is also insufficient explanation in methods or results concerning Geminin treatment.
4. In Figs 3 and 4, the percent RAD51 +/-Geminin + cells considered positive is apparently based on cells having at least 5 foci per cell. If so, the authors need to provide data for at least some representative HRR proficient vs deficient tumors to show the distribution of number of RAD51 foci in Geminin positive nuclei justify the cut off using at least 5 foci per cell.
5. In Fig. 5, the fluorescence data are again difficult to visualize with sufficient clarity to evaluate.

Other Points:

1. Throughout the study, abbreviations are utilized but need to be defined when first used.
2. The small print for tables in the extended data section is difficult to read.

Referee #3 (Comments on Novelty/Model System for Author):

Current limitations of the manuscript, as outlined below, include moderate novelty given that RAD51 has been used as marker for HR proficiency and PARPi sensitivity before, seemingly arbitrary foci detection cutoffs to score "RAD51-positive" vs. "RAD51-negative" cells, and comparably small sample sizes and in some cases (e.g. Fig. 5A) missing statistical analyses.

Referee #3 (Remarks for Author):

PARP inhibitors have become clinically very important reagents and are being used as monotherapy to treat a subset of cancer patients with defects in homologous recombination (HR). There has been a great interest in defining suitable biomarkers, which could predict PARP inhibitor responses and stratify patients according to their expected treatment response. The cellular ability to mount RAD51 foci as homologous recombination intermediate had been suggested before to serve as potential indicator of HR proficiency and thus PARP inhibitor responsiveness. Here, the authors provide evidence from patient-derived tumor xenografts and formalin-fixed paraffin embedded patient samples that RAD51 foci formation can indeed be used to predict PARP inhibitor sensitivity beyond germline BRCA mutations and irrespective of the underlying mechanism of HR-deficiency. An important finding is that steady-state levels of RAD51 foci in patient samples (i.e. without having to induce additional genotoxic stress or DNA damage) could be sufficient to predict PARP inhibitor sensitivity. This has indeed a potential to facilitate patient selection and may thus have high clinical relevance. However, in order to serve as a solid basis for clinical use of the RAD51 score, the experimental data shown should be consolidated and detailed information on foci detection and applied thresholds should be provided. Specific concerns:

- Is the sample size of 18 BC models (4 PARPi sensitive, 14 PARPi resistant), 28 TNBC models, and 24 FFPE tumor samples sufficiently large to substantiate the claims? On several occasions in the manuscript black-and-white discriminations are suggested (e.g. between foci-positive vs. foci-negative samples, PARPi-sensitive vs. PARPi-resistant) and it seems more appropriate to take into considerations that biological responses are often gradual. For instance, PDX270 has low levels of RAD51-positive cells without PARPi treatment (Fig. 3A), the BRCA1 promoter is methylated (Fig. 2A), and the tumor volume shows hardly any increase during treatment (Fig. 1A). Based on this, it could be premature to exclude such cases from PARPi treatment.
- Related to the previous point, it would be informative to provide the initial tumor volume and the end tumor volume in addition to the tumor volume change (Fig. 1A). In addition, would it be possible to include controls in Fig. 1A (e.g. as was done in Fig. 2A)?
- Conversely, are the data available to include the resistant PDX093R in the analysis of Fig. 2A?
- Recently, cases have been reported in which BRCA-deficient cells became resistant to PARPi or cisplatin without restoring RAD51 foci formation (Ray Chaudhuri et al. *Nature*. 2016 Jul 21;535(7612):382-7; Guillemette et al. *Genes Dev*. 2015 Mar 1;29(5):489-94.). The possibility of PARPi-resistance independent of the ability to form RAD51 foci should thus be taken into consideration. Obviously, this would limit the predictive power of the RAD51 score as single biomarker.
- The cutoff at five RAD51 foci per S-phase nucleus seems arbitrary. It would be far better to show absolute foci counts, ideally as scatter plots showing foci counts for each individual S-phase cell (e.g. for Fig. 3A, 4A, 5A). Proper statistical testing should then be performed to assess the significance of differences between the different samples. Information should be provided how a RAD51 focus is defined, i.e. which thresholds are used for foci detection, and how foci detection is affected by the staining quality and image resolution. If the RAD51 score should be used in clinical settings in the future, such information is important to avoid over- or under-sampling of RAD51 foci, which could easily lead to false predictions. To address this point, the authors may want to test themselves how images taken with different microscope objectives and cameras affect the calling of RAD51 foci. Further, how would changing the cutoff of 5 foci to e.g. 3 foci, 8 foci or 10 foci change the result and affect the predictive power of the RAD51 score?
- Similar considerations can be made for gammaH2AX: absolute foci counts should be provided and nuclear intensities of gammaH2AX should be measured as well. Actually, one would expect higher levels of unrepaired DNA damage upon PARPi treatment when the RAD51 response is impaired. This can be seen in Fig. 5A, but not in Fig. 3C and Fig. 4B. How can this discrepancy be explained? Fig. 4B, by the way, is not referred to in the main text.
- Why are three BRCA1-mutated and one BRCA2-mutated samples in Fig. 4A RAD51-proficient? It would be important to know what discriminates these four cases from the other BRCA-mutated samples, which are below the 10% cutoff.
- The quality of the IF images shown in the manuscript has to be improved. Higher magnifications of cells should be shown, the different color channels should be shown as both separate and merged images, and the single cells shown in Fig. 2A and 3A should be complemented by larger fields of views in the supplemental information. Scale bars should be included.
- In Fig. 4A it is not very clear which treatments had been performed on which samples. Perhaps this could be explained better in the text.
- In the introduction, the authors write that DSBs in S and G2 phases of the cell cycle are preferentially repaired by homologous recombination repair (HRR). It would be more correct to write that DSBs in replicated areas of the genome are repaired by HRR (see Saredi et al. *Nature*. 2016 Jun 30;534(7609):714-718 and Pellegrino et al. *Cell Rep*. 2017 May 30;19(9):1819-1831 as references).
- Since the PDX models are likely too time-consuming and the FFPE samples would be more useful to predict PARPi responses in clinical settings, the authors may want to make this point a bit clearer.

Currently, it is not very obvious what the "RAD51 assay feasible in routine tumor samples", mentioned in the title, should be.

- The paper by Oplustilova et al. (Cell Cycle. 2012 Oct 15;11(20):3837-50.) looked at RAD51 as biomarker for PARPi responses and the results reported therein should be discussed, in particular since PARPi sensitivity was observed in cells with normal RAD51 foci formation in this publication.
- The authors may want to discuss also the possibility for ex vivo drug response assays using primary patient material, PARPi and RAD51 foci as readout.

1st Revision - authors' response

09 August 2018

Referee #1 (Comments on Novelty/Model System for Author):

There are some technical concerns noted in my remarks below, but the relevance of a predictive assay, if validated, would be fairly high as this an area of unmet need in the field.

We agree with Reviewer 1 about the technical concerns and we have addressed those that were feasible in this review process.

Referee #1 (Remarks for Author):

The manuscript is seeking to address the need for a better predictive biomarker of PARP response than somatic or germline BRCA1/2 mutations and/or mutational signatures of HR deficiency. RAD51 IHC foci are compared with the Myriad Genetics HRD score, using PDX tumours tested for PARPi sensitivity.

This is an important area of unmet need - better predictive markers of PARP and other HR deficiency exploiting therapies is much needed.

The current manuscript has some problems and falls short in a few areas, weakening the conclusions that can be drawn.

1. Repeatability. The Santa Cruz antibody used here for RAD51, has been discontinued by Santa Cruz, this reviewer believes as part of the Santa Cruz goat shutdown in 2016. This is not the fault of the authors, but it raises the question of repeatability of the present study by others. Perhaps more pertinently it would imply a predictive marker would in any case need to validate other antibodies. One possible remedy would be to re-stain the relevant materials with a currently available RAD51 antibody, to demonstrate reproducibility.

As pointed out by the Referee, the polyclonal Santa Cruz antibody has been discontinued. To address his question, we have validated our findings with a commercial antibody from Abcam. New Figure EV1E shows the correlation between the RAD51 scores using the Santa Cruz antibody and the Abcam antibody on the PDX cohort-1 (N=18), with blinded scoring of the samples by the same reader ($R=0.9630$ and $p<0.0001$). This result demonstrates the repeatability of the RAD51 assay. The results and methods have been amended accordingly to show this data (pages 11 and 21, respectively).

2. The Myriad HRD score is also being discontinued, leading to similar issues over the ability to reproduce or extend the current study.

To the best of our knowledge and after cross-checking with Myriad Genetics Inc. we have confirmed that Myriad's myChoice HRD test has not been discontinued and remains available through Myriad International and Myriad US. The following link summarizes studies/publications of the test regarding Validation, Response and Assay Performance: <https://mychoicehrd.com/physicians/resources/#/response>. In addition, the test is currently included in several clinical trials to further assess the activity of PARPi as maintenance therapy in patients with advanced ovarian cancer (OPINION (NCT03402841), PRIMA (NCT02655016)); as monotherapy in advanced ovarian cancer (QUADRA (NCT02354586), LIGHT (NCT02983799)), in

advanced TNBC (TBB, NCT02401347) or in a variety of tumor types, (NCT02286687), among others.

3. The observations of sensitivity are claimed on the basis of absent RAD51 foci. This is a negative signal and thus carries the risk of technical issues leading to a falsely negative signal and over-estimation of PARP sensitivity. One could pose the result as a positive inverse - ie presence of RAD51 foci predicts resistance. However, using the absence of signal for prediction leading to an active plan to treat, may be unreliable.

We agree with the Referee that an assay based on an absent signal carries the risk of technical issues. However, we have noted that the FDA has recently approved the use of pembrolizumab for the first tissue/site agnostic indication using a PCR test or immunohistochemical tests for MMR-deficiency, the latter being based on absence of signal for prediction (<https://www.fda.gov/drugs/informationondrugs/approveddrugs/ucm560040.htm>).

To avoid carrying the risk of technical issues leading to falsely scoring RAD51-negative samples, we designed several quality controls (QC). Firstly, we used positive and negative external controls with every run of staining. Secondly, we used two internal controls: **a)** the number of geminin-positive cells had to be at least 40, to ensure that there are sufficient cells in S/G2-phase of the cell cycle **b)** the % of γ H2AX/geminin-positive cells had to be at least of 25%, to ensure that the tumor has sufficient endogenous DNA damage. Lack of either geminin- or γ H2AX- positive cells (below those thresholds) would preclude scoring of RAD51. To help understand these scoring criteria, we have added Figure EV1B. After thorough review of these QC, we have excluded one PALB2-related tumor sample from the patient cohort (now Fig 5B) because we could only examine 9 geminin-positive cells and it didn't pass the geminin QC (Pt17 from the first version of the manuscript). In addition, we have included a consort diagram summarizing the attrition factors in the patient sample set (new Fig 5A).

A more favorable situation is scoring RAD51 foci to predict resistance, as shown in our study by Cruz et al (Cruz *et al*, 2018). Future validation of the RAD51 assay in larger clinical cohorts will determine the specificity of the RAD51 assay for both scenarios, i.e. absence of signal for prediction of response to PARPi and presence of signal to predict resistance. In this sense, the RAD51 assay could also be used to establish if homologous recombination repair (HRR)-mutant tumors tested by genetic or genomic tests have restored functional HRR.

Taking into account these considerations and questions number 4 and 12 from Referee 3, we have added the following paragraph to the discussion:

*The RAD51 assay has some limitations. Firstly, when PARPi sensitivity occurs via mechanisms that do not directly impact on the ability of cells to perform HRR, e.g. alterations in ATM (Chen *et al*, 2017; Davies *et al*, 2017; Balmus *et al*, 2018) or in the RNASEH2 complex (Zimmermann *et al*, 2018). Secondly, when PARPi sensitivity occurs via mechanisms that preserve RAD51 foci formation, e.g. alterations in the MRN complex, RAD51API, polymerase eta or ERCC1 (Oplustilova *et al*, 2012; Wiese *et al*, 2007; Kawamoto *et al*, 2005; Postel-Vinay *et al*, 2013). Thirdly, when HRR-deficient tumor have acquired PARPi resistance via RAD51-independent mechanisms that involve replication fork stabilization (Kais *et al*, 2016; Guillemette *et al*, 2015; Chaudhuri *et al*, 2016; Yazinski *et al*, 2017). Fourthly, when a tumor has low proliferation index or low endogenous DNA damage, in which cases the assay would not be feasible.*

4. A more robust comparison to alternative HR detection methods, could include genome wide sequence derived signatures, such as those of HRdetect (Nik-Zainal), which may be more sensitive. The sole reliance on HRD may be a falsely flattering comparison.

We agree with the Referee that genome wide sequence derived signatures may be more sensitive than the HRD score to identify tumors that exhibit DNA repair deficiency by HRR. For this reason, we have recently established collaboration with Prof. Nik-Zainal and results will be part of a future study.

Nonetheless, while genomic scars are highly sensitivity to identify tumors that harbor a DNA repair deficiency by HRR, they are likely to be non-specific. As stated by Watkins and

colleagues: “Tumors whose genome has undergone one or more events that restore high-fidelity homologous recombination are likely to be misclassified as double-strand break repair-deficient and thereby sensitive to PARP inhibitors and DNA damaging chemotherapies as a result of prior repair deficiency and its genomic scarring” (Watkins *et al*, 2014).

Referee #2 (Remarks for Author):

This manuscript compares different approaches with which to identify tumors with functional homologous recombination repair (HRR) defects. This is an important translational research topic, since such tumors are likely to respond to therapy with PARP inhibitors. Moreover, there is clinical evidence these inhibitors have efficacy against tumors beyond those with BRCA1/2 or PALB2 mutations. The design of the study is generally well thought out and the studies well performed. The authors analyzed an initial series of breast tumor PDX models for sensitivity to PARP inhibitors, DNA signatures for HRR defects, genetic aberrations affecting some HRR related genes, and the ability to correlate sensitivity with these or with BRCA1, H2Ax, or RAD51 nuclear foci. The evidence indicated a strong correlation with RAD51 foci so the authors then applied their findings to a test set of PDX tumors and finally to histological sections from patient tumors. Their conclusion is that a low number of otherwise untreated Geminin positive tumor cells with more than 5 RAD51 foci is the best biomarker for HRR deficient tumors/responsive to PARP inhibitors, an assay that can be performed on formalin fixed and paraffin embedded tissue. Unfortunately, the presentation of data needs to be significantly improved particularly with respect to the nuclear staining shown in order to allow meaningful conclusions to be drawn by the reader. Specific points that need to be addressed are as follows:

We thank the Referee for finding the research topic of our manuscript of importance. We have addressed his/her specific concerns as follows.

1. While Fig. 1 shows obvious inhibitory effects of the PARP inhibitor on growth in vivo of 4 tumors, there was at least one other tumor, which grew poorly in response to the inhibitor (Fig. 1B) Was this tumor just a slow grower or did the inhibitor have a significant inhibitory effect. If so, does it also have a functional HRR deficiency as well?

The tumor that grew poorly in response to olaparib in Fig 1B is PDX270 and we have now labeled it. As pointed out by the Referee, PDX270 was just a slow grower in comparison with the other PDX models and it did not exhibit a significant inhibitory effect by olaparib (Appendix Table S1 and Figure for Referees 1).

Regarding the question if PDX270 has a functional HRR deficiency, PDX270 was expected to exhibit functional HRR deficiency due to its *BRCA1* promoter hypermethylation, lack of BRCA1 expression/foci formation and frameshift mutation in *RAD51L*. However, PDX270's RAD51 score was 55% in the olaparib-treated tumors and 13.5% in the untreated tumors, above the 10% cut-off described by Graesser *et al* (Graesser *et al*, 2010) and used in this study. Our data suggests that PDX270 has restored functional HRR by an unknown mechanism and this is consistent with its PARPi-resistance.

2. In Fig.2A, BRCA foci are very difficult to evaluate at the magnification shown. This problem would be worse, if the figure were reduced in size. The results in Fig.2C, D, and E need more detailed explanations since not all readers may be experienced with the assays used in these panels. In Figs. 2D and E, it would be helpful to show the relative expression levels of the exogenous proteins.

We have prepared a new supplementary figure that shows BRCA1 foci in higher magnification and separated channels for four representative models (as requested by Referee 3), shown as Appendix Figure S1. In addition, we have increased the resolution of Figure 2A making the foci easier to evaluate. We have also provided more detailed explanations for Fig 2C, D and E. Figs 2D and E show the relative expression levels of the exogenous proteins.

3. Fig. 3 is very important to the authors' presentation, and the results in the table appear clear. However, it is impossible at the magnification provided for the reader to discern the RAD51 foci in Fig. 3A and also difficult to discern in the panels in which the nuclei are shown at higher magnification (Fig.3B). There is also insufficient explanation in methods or results concerning Geminin treatment.

We have prepared a new supplementary figure to show RAD51 foci with higher magnification (Appendix Figure S2) and increased the resolution of Figure 3A making the foci easier to evaluate. Figure 3B is now shown at a higher magnification and separated channels (as requested by Referee 3). We have also included larger images in Appendix Figures S3 and S4.

Regarding geminin staining, this is a cell proliferation marker that is directly proportional to the cell proliferation index as measured by Ki67 expression (Wohlschlegel *et al*, 2002). Geminin is a master regulator of cell-cycle progression that ensures the timely onset of DNA replication and prevents its rereplication (Ballabeni *et al*, 2013), and used in the literature as counterstaining to mark for S/G2-cell cycle phase (Graeser *et al*, 2010; Naipal *et al*, 2014). We have provided these additional explanations about geminin in the Materials and Methods section (page 22).

4. In Figs 3 and 4, the percent RAD51 +/Geminin + cells considered positive is apparently based on cells having at least 5 foci per cell. If so, the authors need to provide data for at least some representative HRR proficient vs deficient tumors to show the distribution of number of RAD51 foci in Geminin positive nuclei justify the cut off using at least 5 foci per cell.

We have collected data from nine representative tumors that are HRR-proficient vs -deficient, measuring the number of RAD51 foci in 100 geminin-positive cells (Appendix Figure S5A). This data shows that, regarding prediction to PARPi response, the 5 foci-per-cell cut-off is the highest (most stringent) RAD51 foci threshold that provides the most accurate and specific prediction (Appendix Figure S5B-C). Further explanations are provided under question number 5 from Referee 3. This data is now included in the manuscript in page 11 and Appendix Figure S5.

5. In Fig. 5, the fluorescence data are again difficult to visualize with sufficient clarity to evaluate.

To help visualize the fluorescence data in Figure 5 (now in panel C) we show now only tumors from three representative patients, with bigger magnification and higher resolution.

Other Points:

- 1. Throughout the study, abbreviations are utilized but need to be defined when first used.**
- 2. The small print for tables in the extended data section is difficult to read.**

We have reviewed all abbreviations.

Tables in the extended data section are now provided as .docx file with larger font.

Referee #3 (Comments on Novelty/Model System for Author):

Current limitations of the manuscript, as outlined below, include moderate novelty given that RAD51 has been used as marker for HR proficiency and PARPi sensitivity before, seemingly arbitrary foci detection cutoffs to score "RAD51-positive" vs. "RAD51-negative" cells, and comparably small sample sizes and in some cases (e.g. Fig. 5A) missing statistical analyses.

Lack of RAD51 foci formation has indeed been proposed as marker of HRR deficiency and chemotherapy/PARPi sensitivity before (Graeser *et al*, 2010; Naipal *et al*, 2014). However, these studies used an assay that implied the administration of chemotherapy to the patient, or the *ex vivo* irradiation and short-term culture of a fresh tumor sample. The RAD51 assay we have recently proposed for germline *BRCA1/2*-mutant breast cancers (Cruz *et al*, 2018) and for non-*BRCA1/2*-mutant breast cancers in this manuscript is feasible in routine, archival tumor samples, which makes it directly transferrable to the clinic (as this Referee points out below).

The seemingly arbitrary foci detection cut-off issue is being addressed in this revision (see point #5).

The small sample size for *PALB2* pathogenic mutations is inherent to the low frequency of this breast cancer genetic susceptibility, which affects 0.3% of the breast cancer population (Turner, 2017).

The missing statistical analyses have been added (see point #5).

Referee #3 (Remarks for Author):

PARP inhibitors have become clinically very important reagents and are being used as monotherapy to treat a subset of cancer patients with defects in homologous recombination (HR). There has been a great interest in defining suitable biomarkers, which could predict PARP inhibitor responses and stratify patients according to their expected treatment response. The cellular ability to mount RAD51 foci as homologous recombination intermediate had been suggested before to serve as potential indicator of HR proficiency and thus PARP inhibitor responsiveness. Here, the authors provide evidence from patient-derived tumor xenografts and formalin-fixed paraffin embedded patient samples that RAD51 foci formation can indeed be used to predict PARP inhibitor sensitivity beyond germline BRCA mutations and irrespective of the underlying mechanism of HR-deficiency. An important finding is that steady-state levels of RAD51 foci in patient samples (i.e. without having to induce additional genotoxic stress or DNA damage) could be sufficient to predict PARP inhibitor sensitivity. This has indeed a potential to facilitate patient selection and may thus have high clinical relevance. However, in order to serve as a solid basis for clinical use of the RAD51 score, the experimental data shown should be consolidated and detailed information on foci detection and applied thresholds should be provided.

We thank the Referee for finding our data of potentially high clinical relevance. We have addressed his/her specific concerns and explained them below.

Specific concerns:

1. Is the sample size of 18 BC models (4 PARPi sensitive, 14 PARPi resistant), 28 TNBC models, and 24 FFPE tumor samples sufficiently large to substantiate the claims? On several occasions in the manuscript black-and-white discriminations are suggested (e.g. between foci-positive vs. foci-negative samples, PARPi-sensitive vs. PARPi-resistant) and it seems more appropriate to take into considerations that biological responses are often gradual. For instance, PDX270 has low levels of RAD51-positive cells without PARPi treatment (Fig. 3A), the BRCA1 promoter is methylated (Fig. 2A), and the tumor volume shows hardly any increase during treatment (Fig. 1A). Based on this, it could be premature to exclude such cases from PARPi treatment.

- Regarding the sample size calculation, we must acknowledge that our study is exploratory in nature and we worked with as many samples as possible during the study timeframe. The main concern with regards to sample size is when statistical significance is not reached (lack of power). However, the strong associations we describe did reach statistical significance and therefore the sample size is no longer a main concern to substantiate the claims.
- We agree with the Referee that biological responses are often gradual but biomarkers for clinical decisions are ideally dichotomous. In our manuscript, we based the antitumor response categorization on a similar modification of the clinical RECIST criteria applied by Gao *et al* (Therasse *et al*, 2000; Gao *et al*, 2015) and used a RAD51 score cut-off that was previously described (Graeser *et al*, 2010). To address the Referee's point, we have conducted a Pearson correlation analysis for the olaparib response vs. the RAD51 score in the PDX cohort-1. As shown in Figure for Referees 2, there is a significant correlation between the two continuous variables ($p=0.0044$). Obviously, the dispersion of the data amongst the PD models is inherent to the different growth rates of each PDX and therefore the R^2 is relatively low ($R^2=0.4068$). To avoid confusion regarding the significance of the correlation, we suggest to exclusively show the association between the dichotomous variables as we presented in the first version of our manuscript.
- We agree that PDX270 falls on the "intermediate area" (see Figure for Referees 2). As argued by Referee #2, we would like to point out that PDX270 was a slow grower in comparison with the other PDX models and it did not exhibit a significant inhibitory effect

by olaparib (Appendix Table S1 and Figure for Referees 1), which was consistent with its RAD51 score $\geq 10\%$. We agree with the Referee that more data is needed from “intermediate” cases to help refine the RAD51 score cut-off. We expect to obtain them in future retrospective/prospective validations with clinical trial samples from patients having received PARPi-treatment.

2. Related to the previous point, it would be informative to provide the initial tumor volume and the end tumor volume in addition to the tumor volume change (Fig. 1A). In addition, would it be possible to include controls in Fig. 1A (e.g. as was done in Fig. 2A)?

We provide now the initial/end tumor volumes (Appendix Table S1) and as an overlay in Figure for Referees 1.

3. Conversely, are the data available to include the resistant PDX093R in the analysis of Fig. 2A?

PDX93OR was already included in Fig 2A (8th sample from the left).

4. Recently, cases have been reported in which BRCA-deficient cells became resistant to PARPi or cisplatin without restoring RAD51 foci formation (Ray Chaudhuri et al. Nature. 2016 Jul 21;535(7612):382-7; Guillemette et al. Genes Dev. 2015 Mar 1;29(5):489-94.). The possibility of PARPi-resistance independent of the ability to form RAD51 foci should thus be taken into consideration. Obviously, this would limit the predictive power of the RAD51 score as single biomarker.

We have listed the known limitations of the RAD51 assay in the discussion:

The RAD51 assay has some limitations. Firstly, when PARPi sensitivity occurs via mechanisms that do not directly impact on the ability of cells to perform HRR, e.g. alterations in ATM (Chen et al, 2017; Davies et al, 2017; Balmus et al, 2018) or in the RNASEH2 complex (Zimmermann et al, 2018). Secondly, when PARPi sensitivity occurs via mechanisms that preserve RAD51 foci formation, e.g. alterations in the MRN complex, RAD51API, polymerase eta or ERCCI (Oplustilova et al, 2012; Wiese et al, 2007; Kawamoto et al, 2005; Postel-Vinay et al, 2013). Thirdly, when HRR-deficient tumor have acquired PARPi resistance via RAD51-independent mechanisms that involve replication fork stabilization (Kais et al, 2016; Guillemette et al, 2015; Chaudhuri et al, 2016; Yazinski et al, 2017). Fourthly, when a tumor has low proliferation index or low endogenous DNA damage, in which cases the assay would not be feasible.

5. The cutoff at five RAD51 foci per S-phase nucleus seems arbitrary. It would be far better to show absolute foci counts, ideally as scatter plots showing foci counts for each individual S-phase cell (e.g. for Fig. 3A, 4A, 5A). Proper statistical testing should then be performed to assess the significance of differences between the different samples. Information should be provided how a RAD51 focus is defined, i.e. which thresholds are used for foci detection, and how foci detection is affected by the staining quality and image resolution. If the RAD51 score should be used in clinical settings in the future, such information is important to avoid over- or under-sampling of RAD51 foci, which could easily lead to false predictions. To address this point, the authors may want to test themselves how images taken with different microscope objectives and cameras affect the calling of RAD51 foci. Further, how would changing the cutoff of 5 foci to e.g. 3 foci, 8 foci or 10 foci change the result and affect the predictive power of the RAD51 score?

We agree with the Referee that further analysis was needed regarding the RAD51 scoring, since this information is important to avoid over- or under-scoring RAD51 foci that would lead to false predictions. We show now the absolute foci counts as scatter plots and histograms for each individual S-phase cell from nine representative tumors that are HRR-proficient vs -deficient (from Figure 3A). This data shows that, in HRR-deficient tumors, most of the geminin positive cells exhibit zero RAD51 foci (scatter plot shown in Appendix Figure S5A and histogram shown in Figure for Referees 3A). Instead, the HRR-proficient cells exhibit a bimodal distribution of RAD51 foci with the high-RAD51 foci distribution peaking at 5-foci-per-cell (Figure for Referees 3B). Regarding prediction to PARPi response, we demonstrate that the 5-foci-per-cell cut-off is the

highest (most stringent) threshold that provides the most accurate and specific prediction (Appendix Figure S5B-C). This data is now included in the manuscript in page 11 and Appendix Figure S5.

Regarding proper statistical testing, we calculated:

- (a) The area under the ROC curve (AUC) to estimate the prediction capacity of the RAD51 score PARPi-response in Figure 3A (AUC = 1) and Figure 4A (AUC = 1). The CI95% for ROC AUC is not reported because when AUC = 1 the CI is always (1-1) and can be misleading.
- (b) A logistic regression model (odds ratio, OR) to estimate the association between carrying a *PALB2*-mutation and having a low RAD51 score for Figure 5A, now Figure 5B (OR=62.4, $p=0.0003$)

In addition, we have added a Bootstrap statistical test to compare the ROC curves between the RAD51 score and the HRD score, concluding that in PDX cohort-2 the prediction capacity of the RAD51 score was higher than the HRD score (now Fig 4B, difference between AUC = 0.27 (0.08-0.46) $p=0.005$).

As per the RAD51 focus, we would like to clarify that RAD51 foci quantification was performed on live images and it was therefore not affected by the digital image resolution. RAD51 foci were seen as bright nuclear spots of mean diameter of 0.66 μm (ranging 0.42-1.15) and mean area of 0.37 μm^2 (0.14-1.04). They could be clearly distinguished from homogeneous (background) and non-homogeneous noise. Staining quality in primary tumors from patients with HBOC syndrome was generally not an issue, with the exception of two tumors (out of 29 samples) that were not well preserved and other two that had insufficient geminin-positive cells (summarized in new Fig 5A).

In order to validate the RAD51 assay in samples from clinical trials and to implement it as a diagnostic test several steps need to be accomplished. We maintained strict scoring criteria and always included positive and negative control samples to help confirm the validity and reproducibility of the results across experiments (new Fig EV1B). Another important point, as observed by the Referee, is to standardize the assay to be used on digital images taken with different microscopes and cameras. In collaboration with Prof. Carsten Denkert (Charité, Berlin), we are currently setting up the image digitalization of RAD51-stained tissue microarrays. This is a key step to move on with the analysis of clinical trial tumor samples. Moreover, since manual counting is time-consuming, we are considering to automate foci quantification with available software (e.g. FoCo, CellProfiler or ImageJ) or a customized one (if needed). Nevertheless, manual foci count is still considered to be the gold standard and it is regularly used as a benchmark to validate the performance of automatic methods.

We thank the Referee for his/her insight as this review process has been very helpful for the future validation/implementation of the RAD51 assay in the clinic.

6. Similar considerations can be made for gammaH2AX: absolute foci counts should be provided and nuclear intensities of gammaH2AX should be measured as well. Actually, one would expect higher levels of unrepaired DNA damage upon PARPi treatment when the RAD51 response is impaired. This can be seen in Fig. 5A, but not in Fig. 3C and Fig. 4B. How can this discrepancy be explained? Fig. 4B, by the way, is not referred to in the main text.

In our study, γH2AX is used as quality check for DNA damage and not for scoring HRR deficiency. γH2AX foci is a widely-used marker of DNA double strand breaks. Therefore, we argue that scoring absolute foci counts and foci intensity is unnecessary.

We apologize if the γH2AX data has led to confusion. We only observed significantly higher levels of γH2AX /geminin-positive cells in RAD51-deficient vs. RAD51-proficient tumors in vehicle-treated PDXs from cohort-1 (Fig 3C). However, DNA-damage induction upon olaparib treatment did not result in higher levels of γH2AX /geminin-positive cells in RAD51-deficient compared to RAD51-proficient (Fig 3C). These seemingly discrepant results may be due to the low sample size to address this question. Nonetheless, when combining all our data with the 13 PDXs published by Cruz et al, these differences were not statistically significant, neither in vehicle- nor olaparib-treated PDXs (Figure for Referees 4A). Finally, different levels of γH2AX /geminin-positive cells were not

observed in cohort-2 (Fig 4B (now Figure for Referees 4B)) nor in the patient cohort (Fig 5A (now Fig 5B), statistics in Figure for Referees 4C).

The lack of observation of higher baseline/induced DNA damage in HRR-deficient tumors might be the result of various factors. Firstly, we quantified γ H2AX foci in geminin-positive cells (S/G2-cell cycle phase) and not in the overall population. It might be that HRR-deficient cells carry higher levels of unrepaired DNA damage (with/without PARPi treatment) and this is noted on the overall population. Second, it might also be that HRR-deficient cells with supra-physiological DNA damage undergo cell death with/without PARPi which might be better visualized by pan-nuclear γ H2AX than with γ H2AX foci. In summary, we don't think that there is discrepancy in our data and there might be better readouts to analyze this hypothesis. To avoid confusion, we have removed Fig 4B and the statistical analysis between baseline tumors in Fig 3C.

7. Why are three BRCA1-mutated and one BRCA2-mutated samples in Fig. 4A RAD51-proficient? It would be important to know what discriminates these four cases from the other BRCA-mutated samples, which are below the 10% cutoff.

There are three *BRCA1*-mutated and one *BRCA2*-mutated PDXs that exhibit RAD51 foci and are PARPi-resistant. We have investigated the known mechanisms of PARPi resistance in these models with the available exome sequencing (Seq) data and performed immunofluorescence analyses. Exome Seq data did not identify any alteration in known PARPi-resistant genes. Loss of 53BP1 did not explain PARPi resistance in the three *BRCA1*-mutated PDXs that exhibited RAD51 foci (new Fig 4C). Instead, two of them exhibited BRCA1 foci by immunofluorescence, indicative of functional HRR restoration by BRCA1 hypomorphic variants as previously described (Cruz *et al*, 2018). We have added the aforementioned 53BP1/BRCA1 statements to the results section (page 12 and new Fig 4C).

8. The quality of the IF images shown in the manuscript has to be improved. Higher magnifications of cells should be shown, the different color channels should be shown as both separate and merged images, and the single cells shown in Fig. 2A and 3A should be complemented by larger fields of views in the supplemental information. Scale bars should be included.

We apologize for the quality of the IF images on the previous version of the manuscript. We have now increased the image resolution from Figs. 2A and 3A and show higher magnification of cells. The different color channels are shown as separate and merged images for representative models from Fig 2A and 3A (Appendix Figures S1 and S2, respectively) and for Fig 3B as main figure and Appendix Figures S3 and S4. Scale bars are also included.

9. In Fig. 4A it is not very clear which treatments had been performed on which samples. Perhaps this could be explained better in the text.

We have amended the text and the figure legend to make the treatments and doses clearer.

10. In the introduction, the authors write that DSBs in S and G2 phases of the cell cycle are preferentially repaired by homologous recombination repair (HRR). It would be more correct to write that DSBs in replicated areas of the genome are repaired by HRR (see Saredi et al. Nature. 2016 Jun 30;534(7609):714-718 and Pellegrino et al. Cell Rep. 2017 May 30;19(9):1819-1831 as references).

We have amended the text and added the two citations.

11. Since the PDX models are likely too time-consuming and the FFPE samples would be more useful to predict PARPi responses in clinical settings, the authors may want to make this point a bit clearer. Currently, it is not very obvious what the "RAD51 assay feasible in routine tumor samples", mentioned in the title, should be.

We have strengthened this point in the first sentence of the last paragraph of the Discussion (page 15): *In summary, we demonstrate that the RAD51 assay is feasible in routine FFPE tumor samples and could identify several populations that might be sensitive to PARPi.*

12. The paper by Oplustilova et al. (Cell Cycle. 2012 Oct 15;11(20):3837-50.) looked at RAD51 as biomarker for PARPi responses and the results reported therein should be discussed, in particular since PARPi sensitivity was observed in cells with normal RAD51 foci formation in this publication.

As we have mentioned in point #4, PARPi synthetic lethality may occur via RAD51-independent mechanisms, and these do not result in lack of RAD51 foci formation.

13. The authors may want to discuss also the possibility for ex vivo drug response assays using primary patient material, PARPi and RAD51 foci as readout.

We thank the reviewer for this suggestion. *Ex vivo* drug response assays using primary patient explants are a potentially valuable tool to predict response to PARPi vs. other drugs. This might be especially relevant given the low take rate and slow growth rate of the breast cancer PDXs. Nonetheless, in this study we would like to emphasize the sensitivity and specificity of the RAD51 assay to predict PARPi response beyond the germline BRCA1/2 condition and its feasibility on routine FFPE tumor samples, which is of clinical relevance since a fresh sample would not be required.

REFERENCES

- Ballabeni A, Zamponi R, Moore JK, Helin K & Kirschner MW (2013) Geminin deploys multiple mechanisms to regulate Cdt1 before cell division thus ensuring the proper execution of DNA replication. *Proc. Natl. Acad. Sci.* **110**: E2848–E2853
- Balmus G, Pilger D, Coates J, Demir M, Sczaniecka-Clift M, Barros A, Woods M, Fu B, Yang F, Chen E, Ostermaier M, Stankovic T, Ponstingl H, Herzog M, Yusa K, Munoz-Martinez F, Durant ST, Galanty Y, Beli P, Adams DJ, et al (2018) ATM orchestrates the DNA-damage response to counter toxic non-homologous end-joining at broken replication forks. *bioRxiv* doi: 10.1101/330043 [PREPRINT]
- Chaudhuri RA, Callen E, Ding X, Gogola E, Duarte AA, Lee J-E, Wong N, Lafarga V, Calvo JA, Panzarino NJ, John S, Day A, Crespo AV, Shen B, Starnes LM, Rutter JR de, Daniel JA, Konstantinopoulos PA, Cortez D, Cantor SB, et al (2016) Replication fork stability confers chemoresistance in BRCA-deficient cells. *Nature* **535**: 382–387
- Chen C-C, Kass EM, Yen W-F, Ludwig T, Moynahan ME, Chaudhuri J & Jasin M (2017) ATM loss leads to synthetic lethality in BRCA1 BRCT mutant mice associated with exacerbated defects in homology-directed repair. *Proc. Natl. Acad. Sci.* **114**: 7665–7670
- Cruz C, Castroviejo-Bermejo M, Gutiérrez-Enriquez S, Llop-Guevara A, Ibrahim YH, Gris-Oliver A, Bonache S, Moracho B, Bruna A, Rueda OM, Lai Z, Polanska UM, Jones GN, Kristel P, de Bustos L, Guzman M, Rodriguez O, Grueso J, Montalban G, Caratú G, et al (2018) RAD51 foci as a functional biomarker of homologous recombination repair and PARP inhibitor resistance in germline BRCA-mutated breast cancer. *Ann. Oncol.* **29**: 1203–1210
- Davies H, Glodzik D, Morganella S, Yates LR, Staaf J, Zou X, Ramakrishna M, Martin S, Boyault S, Sieuwerts AM, Simpson PT, King TA, Raine K, Eyfjord JE, Kong G, Borg Å, Birney E, Stunnenberg HG, van de Vijver MJ, Børresen-Dale A-L, et al (2017) HRDetect is a predictor of BRCA1 and BRCA2 deficiency based on mutational signatures. *Nat. Med.* **23**: 517–525
- Gao H, Korn JM, Ferretti S, Monahan JE, Wang Y, Singh M, Zhang C, Schnell C, Yang G, Zhang Y, Balbin OA, Barbe S, Cai H, Casey F, Chatterjee S, Chiang DY, Chuai S, Cogan SM, Collins SD, Dammassa E, et al (2015) High-throughput screening using patient-derived tumor xenografts to predict clinical trial drug response. *Nat. Med.* **21**: 1318–25
- Graeser M, McCarthy A, Lord CJ, Savage K, Hills M, Salter J, Orr N, Parton M, Smith IE, Reis-Filho JS, Dowsett M, Ashworth A & Turner NC (2010) A marker of homologous recombination predicts pathologic complete response to neoadjuvant chemotherapy in primary breast cancer. *Clin. Cancer Res.* **16**: 6159–6168
- Guillemette S, Serra RW, Peng M, Hayes JA, Konstantinopoulos PA, Green MR & Cantor SB (2015) Resistance to therapy in BRCA2 mutant cells due to loss of the nucleosome remodeling factor CHD4. *Genes Dev.* **29**: 489–94
- Kais Z, Rondinelli B, Holmes A, O’Leary C, Kozono D, D’Andrea AD & Ceccaldi R (2016) FANCD2 Maintains Fork Stability in BRCA1/2-Deficient Tumors and Promotes Alternative End-Joining DNA Repair. *Cell Rep.* **15**: 2488–2499

- Kawamoto T, Araki K, Sonoda E, Yamashita YM, Harada K, Kikuchi K, Masutani C, Hanaoka F, Nozaki K, Hashimoto N & Takeda S (2005) Dual Roles for DNA Polymerase η in Homologous DNA Recombination and Translesion DNA Synthesis. *Mol. Cell* **20**: 793–799
- Naipal KAT, Verkaik NS, Ameziane N, van Deurzen CHM, ter Brugge P, Meijers M, Sieuwerts AM, Martens JW, O'Connor MJ, Vrieling H, Hoeijmakers JHJ, Jonkers J, Kanaar R, de Winter JP, Vreeswijk MP, Jager A & van Gent DC (2014) Functional Ex Vivo Assay to Select Homologous Recombination-Deficient Breast Tumors for PARP Inhibitor Treatment. *Clin. Cancer Res.* **20**: 4816–4826
- Oplustilova L, Wolanin K, Mistrik M, Korinkova G, Simkova D, Bouchal J, Lenobel R, Bartkova J, Lau A, O'Connor MJ, Lukas J & Bartek J (2012) Evaluation of candidate biomarkers to predict cancer cell sensitivity or resistance to PARP-1 inhibitor treatment. *Cell Cycle* **11**: 3837–3850
- Postel-Vinay S, Bajrami I, Friboulet L, Elliott R, Fontebasso Y, Dorvault N, Olaussen KA, André F, Soria J-C, Lord CJ & Ashworth A (2013) A high-throughput screen identifies PARP1/2 inhibitors as a potential therapy for ERCC1-deficient non-small cell lung cancer. *Oncogene* **32**: 5377–5387
- Therasse P, Arbuuck SG, Eisenhauer EA, Wanders J, Kaplan RS, Rubinstein L, Verweij J, Van Glabbeke M, van Oosterom AT, Christian MC & Gwyther SG (2000) New guidelines to evaluate the response to treatment in solid tumors. European Organization for Research and Treatment of Cancer, National Cancer Institute of the United States, National Cancer Institute of Canada. *J. Natl. Cancer Inst.* **92**: 205–16
- Turner NC (2017) Signatures of DNA-Repair Deficiencies in Breast Cancer. *N. Engl. J. Med.* **377**: 2490–2492
- Watkins JA, Irshad S, Grigoriadis A & Tutt AN (2014) Genomic scars as biomarkers of homologous recombination deficiency and drug response in breast and ovarian cancers. *Breast Cancer Res.* **16**: 211
- Wiese C, Dray E, Groesser T, San Filippo J, Shi I, Collins DW, Tsai M-S, Williams GJ, Rydberg B, Sung P & Schild D (2007) Promotion of Homologous Recombination and Genomic Stability by RAD51AP1 via RAD51 Recombinase Enhancement. *Mol. Cell* **28**: 482–490
- Wohlschlegel JA, Kutok JL, Weng AP & Dutta A (2002) Expression of Geminin as a Marker of Cell Proliferation in Normal Tissues and Malignancies. *Am. J. Pathol.* **161**: 267–273
- Yazinski SA, Comaills V, Buisson R, Genoie M-M, Nguyen HD, Ho CK, Todorova Kwan T, Morris R, Lauffer S, Nussenzweig A, Ramaswamy S, Benes CH, Haber DA, Maheswaran S, Birrer MJ & Zou L (2017) ATR inhibition disrupts rewired homologous recombination and fork protection pathways in PARP inhibitor-resistant BRCA-deficient cancer cells. *Genes Dev.* **31**: 318–332
- Zimmermann M, Murina O, Reijns MAM, Agathangelou A, Challis R, Tarnauskaite Ž, Muir M, Fluteau A, Aregger M, McEwan A, Yuan W, Clarke M, Lambros MB, Paneesha S, Moss P, Chandrashekhar M, Angers S, Moffat J, Brunton VG, Hart T, et al (2018) CRISPR screens identify genomic ribonucleotides as a source of PARP-trapping lesions. *Nature* **559**: 285–289

Figure for Referees 1

Figure for Referees 1. Tumor volume change (%) in PDXs from cohort-1 following vehicle- or olaparib-treatment. Graph showing the percentage of tumor volume change in vehicle- (grey dots) and olaparib-treated (green bars) tumors compared to the tumor volume on day 1. Tumor Volume Change of +20% and -30% are marked by dotted lines to indicate the range of PR, SD and PD. The box underneath summarizes different characteristics of each model and the clinical context at the moment of PDX implantation. TNBC, Triple Negative Breast Cancer; ER+BC, Estrogen Receptor positive Breast Cancer; P, primary; M, metastasis. Error bars indicate SEM from independent tumors.

Figure for Referees 2

Figure for Referees 2. The RAD51 score correlates with the percentage of Tumor Volume Change following olaparib-treatment. Pearson correlation between the RAD51 score (percentage of RAD51 (+) / Geminin (+) cells, assessed in untreated FFPE tumor samples) and the Tumor Volume Change in olaparib-treated tumors from PDX cohort-1. Each dot represents one PDX model. Error bars indicate SEM from independent tumors treated with olaparib. The table shows the statistical analysis.

Figure for Referees 3

A

B

Figure for Referees 3. Distribution of the number of RAD51 foci-per-cell in PARPi-sensitive and PARPi-resistant PDXs from cohort-1. Histogram representing the number geminin-positive cells with increasing number of RAD51 foci (0 to 50) in **A)** three PARPi-sensitive and **B)** six PARPi-resistant PDXs from cohort-1. Each color represents one PDX model.

Figure for Referees 4

Figure for Referees 4. HRR-deficient tumors did not show higher levels of DNA-damage in S/G2-phase of the cell cycle. Quantification of geminin-positive cells that exhibit gH2AX foci in RAD51 high vs RAD51 low tumors from **A)** PDX cohort-1 plus 13 additional gBRCA PDXs, **B)** PDX cohort-2 and **C)** tumor from patients with HBOC syndrome. Unpaired t-test in CR/PR vs PD tumors and in tumors with high vs low RAD51 score. Paired t-test in vehicle- vs PARPi-treated tumors.

Thank you for the submission of your revised manuscript to EMBO Molecular Medicine. We have now received the enclosed reports from the referees that were asked to re-assess it. As you will see

the reviewers are now globally supportive and I am pleased to inform you that we will be able to accept your manuscript pending minor editorial amendments, including a response to Referee #3.

***** Reviewer's comments *****

Referee #2 (Comments on Novelty/Model System for Author):

The authors have made substantial improvements in the revised manuscript in response to the reviewers' comments

Referee #2 (Remarks for Author):

The authors have made substantial improvements in the revised manuscript in response to the reviewers' comments. Thus, I recommend acceptance of the revised manuscript.

Referee #3 (Remarks for Author):

The authors have addressed many of my initial concerns and now provide images of better resolution as well as additional analyses to corroborate their claims. The following couple of points, however, may deserve additional attention:

- 1) In the May 2018 issue of *Annals of Oncology* the authors published a related study on germline BRCA-mutated breast cancer (Cruz et al. RAD51 foci as a functional biomarker of homologous recombination repair and PARP inhibitor resistance in germline BRCA-mutated breast cancer. *Annals of Oncology*, Volume 29, Issue 5, 1 May 2018, Pages 1203-1210), which in the initial submission of the current work was cited as „accepted" manuscript. From a conceptual point of view the two articles are quite related and reach very similar conclusions. If this, from an editorial perspective, does not affect the novelty of the current work, it might nevertheless be more appropriate to refer to and discuss the Cruz et al. paper earlier in the current manuscript, e.g. already in the introduction.
- 2) The new histograms provide valuable information. Would the bimodal distribution in the PARPi-resistant samples indicate that sub-clones within these samples might be sensitive to PARPi (i.e. the geminin-positive ones with less than 5 RAD51 foci in Figure 3 for the Referees; and the ones with less than 5 RAD51 foci in Appendix Figure S5)? This should be discussed.
- 3) In Figure 3A, should one be able to appreciate an increase in RAD51 foci number upon Olaparib treatment, e.g. as can be seen with cell lines? Or is it merely the percentage of RAD51 (+) cells that changes upon treatment?
- 4) Related to the previous points, wouldn't it make sense to complement Appendix Figure S5 with the foci counts after Olaparib treatment?
- 5) The authors may want to consider adding the informative Figure 2 for the Referees to the manuscript.
- 6) On page 15, the authors should define better what they mean by "in real time". Please be more specific and add information on how long it takes from sample collection to assay result.
- 7) Since gammaH2AX may well be a good biomarker for PARP inhibitor sensitivity itself (Redon et al., *Clin Cancer Res.* 2010 Sep 15;16(18):4532-42) and could be combined with the RAD51 assay, the potential prognostic value of gammaH2AX should be discussed in a more differentiated manner.
- 8) The discussion on limitations of the RAD51 assay (page15) is very good and an important addition. In the meantime, mutations in PARP1 and loss of PARG have also been shown to confer PARPi resistance (Pettitt et al. *Nat Commun.* 2018 May 10;9(1):1849; Gogola et al. *Cancer Cell.* 2018 Jun 11;33(6):1078-1093; Michelena et al. *Nat Commun.* 2018 Jul 11;9(1):2678), as well as loss of 53BP1 and the associated "Shieldin" complex (summarized by Greenberg *Nat Cell Biol.*

2018 Aug;20(8):862-863), and I would recommend to extend the discussion to cover also these examples of PARPi resistance.

2nd Revision - authors' response

19 September 2018

Referee #3 (Remarks for Author):

The authors have addressed many of my initial concerns and now provide images of better resolution as well as additional analyses to corroborate their claims. The following couple of points, however, may deserve additional attention:

1) In the May 2018 issue of Annals of Oncology the authors published a related study on germline BRCA-mutated breast cancer (Cruz et al. RAD51 foci as a functional biomarker of homologous recombination repair and PARP inhibitor resistance in germline BRCA-mutated breast cancer. *Annals of Oncology*, Volume 29, Issue 5, 1 May 2018, Pages 1203-1210), which in the initial submission of the current work was cited as „accepted" manuscript. From a conceptual point of view the two articles are quite related and reach very similar conclusions. If this, from an editorial perspective, does not affect the novelty of the current work, it might nevertheless be more appropriate to refer to and discuss the Cruz et al. paper earlier in the current manuscript, e.g. already in the introduction.

We have added a reference to our previous work on BRCA-mutated breast cancer in the introduction:

Other approaches entail the quantification of BRCA1 promoter methylation, BRCA1 mRNA expression or the detection of the HRR protein RAD51 forming nuclear foci after DNA damage, as surrogate of HRR functionality (ter Brugge et al, 2016; Graeser et al, 2010; Naipal et al, 2014). In this sense, we showed that, in gBRCA tumors, RAD51 foci could be detected in untreated samples and correlated with PARPi resistance regardless of the underlying mechanism restoring HRR function (Cruz et al, 2018).

2) The new histograms provide valuable information. Would the bimodal distribution in the PARPi-resistant samples indicate that sub-clones within these samples might be sensitive to PARPi (i.e. the geminin-positive ones with less than 5 RAD51 foci in Figure 3 for the Referees; and the ones with less than 5 RAD51 foci in Appendix Figure S5)? This should be discussed.

We thank the Referee for finding the histograms informative. We would argue that RAD51-negative cells within PARPi-resistant samples could indicate:

- a) HRR-deficient, PARPi-sensitive sub-clones that will be killed by the drug, as pointed by the Referee, but will be overgrown by the PARPi-resistant tumour cells
- b) tumour cells with low or absent DNA damage that have not engaged HRR in S/G2-phase of the cell cycle

In this sense, we are currently working to improve the immunofluorescence assay to enable triple staining of RAD51, γ H2AX and geminin, which will help distinguish the above-mentioned scenarios.

3) In Figure 3A, should one be able to appreciate an increase in RAD51 foci number upon Olaparib treatment, e.g. as can be seen with cell lines? Or is it merely the percentage of RAD51 (+) cells that changes upon treatment?

We do not appreciate a marked increase in RAD51 foci number upon olaparib treatment for most of the PDXs studied, except for PDX94 and PDX270 that show few cells with higher RAD51 foci (Appendix Fig S5B).

4) Related to the previous points, wouldn't it make sense to complement Appendix Figure S5 with the foci counts after Olaparib treatment?

As requested by the Referee, we have complemented Appendix Figure S5B with the foci counts after olaparib treatment.

5) The authors may want to consider adding the informative Figure 2 for the Referees to the manuscript.

We have added the linear correlation between the RAD51 score and the % of Tumor Volume Change as supplementary figure Fig EV1D.

6) On page 15, the authors should define better what they mean by "in real time". Please be more specific and add information on how long it takes from sample collection to assay result.

We have reformulated the sentence as: “*This functional assay provides an accurate measurement of HRR status and PARPi sensitivity at the time of treatment decision-making.*”

7) Since gammaH2AX may well be a good biomarker for PARP inhibitor sensitivity itself (Redon et al., Clin Cancer Res. 2010 Sep 15;16(18):4532-42) and could be combined with the RAD51 assay, the potential prognostic value of gammaH2AX should be discussed in a more differentiated manner.

Redon and co-workers review the role of γ H2AX in the DNA damage response. Specifically, they claim:

- (a) its utility as pharmacodynamic biomarker, for various chemotherapies and PARP inhibitors
- (b) that γ H2AX does not seem to correlate with treatment response

The pharmacodynamic effect of olaparib on γ H2AX is indeed observed in PDX cohort-1 (Figure 3C). However, as explained in Referee’s 3 question 6 (from the previous review), PARPi-sensitive tumors do not exhibit higher levels of DNA damage measured with the γ H2AX biomarker.

Finally, we agree with the Referee that γ H2AX will be helpful once incorporated in a multiplexed assay, but as stated before in question number 2, to provide the quality check of DNA damage within the same slide/tumor cell.

8) The discussion on limitations of the RAD51 assay (page15) is very good and an important addition. In the meantime, mutations in PARP1 and loss of PARG have also been shown to confer PARPi resistance (Pettitt et al. Nat Commun. 2018 May 10;9(1):1849; Gogola et al. Cancer Cell. 2018 Jun 11;33(6):1078-1093; Michelena et al. Nat Commun. 2018 Jul 11;9(1):2678), as well as loss of 53BP1 and the associated "Shieldin" complex (summarized by Greenberg Nat Cell Biol. 2018 Aug;20(8):862-863), and I would recommend to extend the discussion to cover also these examples of PARPi resistance.

We have added two mechanisms of resistance to PARPi that will not be captured by the RAD51 assay: the mutations in PARP1 and loss of PARG. On the contrary, loss of 53BP1 or the “Shieldin” complex members would result in a RAD51-dependent restoration of HRR and not represent a limitation for the RAD51 assay.

This is the proposed final text:

The RAD51 assay has some limitations. Firstly, when PARPi sensitivity occurs via mechanisms that do not directly impact on the ability of cells to perform HRR, e.g. alterations in ATM (Chen et al, 2017; Davies et al, 2017; Balmus et al, 2018) or in the RNASEH2 complex (Zimmermann et al, 2018). Secondly, when PARPi sensitivity occurs via mechanisms that preserve RAD51 foci formation, e.g. alterations in the MRN complex, RAD51AP1, polymerase eta or ERCC1 (Oplustilova et al, 2012; Wiese et al, 2007; Kawamoto et al, 2005; Postel-Vinay et al, 2013) . Thirdly, when HRR-deficient tumor have acquired PARPi resistance via RAD51-independent mechanisms such as loss of PARG, mutations in PARP1 or those that involve replication fork stabilization (Gogola et al, 2018; Michelena et al, 2018; Pettitt et al, 2018; Kais et al, 2016; Guillemette et al, 2015; Chaudhuri et al, 2016; Yazinski et al, 2017). Fourthly, when a tumor has low proliferation index or low endogenous DNA damage, in which cases the assay would not be feasible.

Corresponding Author Name: Violeta Serra & Judith Balmaña

Manuscript Number: EMM-2018-09172-V2